# How to Find Your Friendly Neighborhood: Graph Attention Design with Self-Supervision

**Dongkwan Kim & Alice Oh**
KAIST, Republic of Korea
dongkwan.kim@kaist.ac.kr, alice.oh@kaist.edu

## Abstract

Attention mechanism in graph neural networks is designed to assign larger weights to important neighbor nodes for better representation. However, what graph attention learns is not understood well, particularly when graphs are noisy. In this paper, we propose a self-supervised graph attention network (SuperGAT), an improved graph attention model for noisy graphs. Specifically, we exploit two attention forms compatible with a self-supervised task to predict edges, whose presence and absence contain the inherent information about the importance of the relationships between nodes. By encoding edges, SuperGAT learns more expressive attention in distinguishing mislinked neighbors. We find two graph characteristics influence the effectiveness of attention forms and self-supervision: homophily and average degree. Thus, our recipe provides guidance on which attention design to use when those two graph characteristics are known. Our experiment on 17 real-world datasets demonstrates that our recipe generalizes across 15 datasets of them, and our models designed by recipe show improved performance over baselines.

## 1 Introduction

Graphs are widely used in various domains, such as social networks, biology, and chemistry. Since their patterns are complex and irregular, learning to represent graphs is challenging (Bruna et al., 2014; Henaff et al., 2015; Defferrard et al., 2016; Duvenaud et al., 2015; Atwood & Towsley, 2016). Recently, graph neural networks (GNNs) have shown a significant performance improvement by generating features of the center node by aggregating those of its neighbors (Zhou et al., 2018; Wu et al., 2020). However, real-world graphs are often noisy with connections between unrelated nodes, and this causes GNNs to learn suboptimal representations. Graph attention networks (GATs) (Veličković et al., 2018) adopt self-attention to alleviate this issue. Similar to attention in sequential data (Luong et al., 2015; Bahdanau et al., 2015; Vaswani et al., 2017), graph attention captures the *relational importance of a graph*, in other words, the degree of importance of each of the neighbors to represent the center node. GATs have shown performance improvements in node classification, but they are inconsistent in the degree of improvement across datasets, and there is little understanding of what graph attention actually learns.

Hence, there is still room for graph attention to improve, and we start by assessing and learning the relational importance for each graph via self-supervised attention. We leverage edges that explicitly encode information about the importance of relations provided by a graph. If node $i$ and $j$ are linked, they are more relevant to each other than others, and if node $i$ and $j$ are not linked, they are not important to each other. Although conventional attention is trained without direct supervision, if we have prior knowledge about what to attend, we can supervise attention using them (Knyazev et al., 2019; Yu et al., 2017). Specifically, we exploit a self-supervised task, using the attention value as input to predict the likelihood that an edge exists between nodes.

To encode edges in graph attention, we first analyze what graph attention learns and how it relates to the presence of edges. In this analysis, we focus on two commonly used attention mechanisms, GAT's original single-layer neural network (GO) and dot-product (DP), as building blocks of our proposed model, self-supervised graph attention network (SuperGAT). We observe that DP attention shows better performance than GO attention in the task to predict link with attention value. On the other hand, GO attention outperforms DP attention in capturing label-agreement between a target

node and its neighbors. Based on our analysis, we propose two variants of SuperGAT, scaled dot-product (SD) and mixed GO and DP (MX), to emphasize the strength of GO and DP.

Then, which graph attention models the relational importance best and produces the best node representations? We find that it depends on the average degree and homophily of the graph. We generate synthetic graph datasets with various degrees and homophily, and analyze how the choice of attention affects node classification performance. Based on this result, we propose a recipe to design graph attention with edge self-supervision that works most effectively for given graph characteristics. We conduct experiments on a total of 17 real-world datasets and demonstrate that our recipe can be generalized across them. In addition, we show that models developed by our method improve performance over baselines.

We present the following contributions. First, we present models with self-supervised attention using edge information. Second, we analyze the classic attention forms GO and DP using label-agreement and link prediction tasks, and this analysis reveals that GO is better at label agreement and DP at link prediction. Third, we propose recipes to design graph attention concerning homophily and average degree and confirm its validity through experiments on real-world datasets. We make our code available for future research (`https://github.com/dongkwan-kim/SuperGAT`).

## 2 RELATED WORK

Deep neural networks are actively studied in modeling graphs, for example the graph convolutional networks (Kipf & Welling, 2017) which approximate spectral graph convolution (Bruna et al., 2014; Defferrard et al., 2016). A representative work in a non-spectral way is the graph attention networks (GATs) (Veličković et al., 2018) which model relations in graphs using self-attention mechanism (Vaswani et al., 2017). Similar to attention in sequence data (Bahdanau et al., 2015; Luong et al., 2015; Vaswani et al., 2017), variants of attention in graph neural networks (Thekumparampil et al., 2018; Zhang et al., 2018; Wang et al., 2019a; Gao & Ji, 2019; Zhang et al., 2020; Hou et al., 2020) are trained without direct supervision. Our work is motivated by studies that improve attention's expressive power by giving direct supervision (Knyazev et al., 2019; Yu et al., 2017). Specifically, we employ a self-supervised task to predict edge presence from attention value. This is in line with two branches of recent GNN research: self-supervision and graph structure learning.

Recent studies about self-supervised learning for GNNs propose tasks leveraging the inherent information in the graph structure: clustering, partitioning, context prediction after node masking, and completion after attribute masking (Hu et al., 2020b; Hui et al., 2020; Sun et al., 2020; You et al., 2020). To the best of our knowledge, ours is the first study to analyze self-supervised learning of graph attention with edge information. Our self-supervised task is similar to link prediction (Liben-Nowell & Kleinberg, 2007), which is a well-studied problem and recently tackled by neural networks (Zhang & Chen, 2017; 2018). Our DP attention to predict links is motivated by graph autoencoder (GAE) (Kipf & Welling, 2016) and its extensions (Pan et al., 2018; Park et al., 2019) reconstructing edges by applying a dot-product decoder to node representations.

Graph structure learning is an approach to learn the underlying graph structure while jointly learning downstream tasks (Jiang et al., 2019; Franceschi et al., 2019; Klicpera et al., 2019; Stretcu et al., 2019; Zheng et al., 2020). Since real-world graphs often have noisy edges, encoding structure information contributes to learn better representation. However, recent models with graph structure learning suffer from high memory and computational complexity. Some studies target all spaces where edges can exist, so they require $O(|V|^2)$ space and computational complexity (Jiang et al., 2019; Franceschi et al., 2019). Others using iterative training (or co-training) between the GNNs and the structure learning model are time-intensive in training (Franceschi et al., 2019; Stretcu et al., 2019). We moderate this problem using graph attention, which consists of parallelizable operations, and our model is built on it without additional parameters. Our model learns attention values that are predictive of edges, and this can be seen as a new paradigm of learning the graph structure.

## 3 MODEL

In this section, we review the original GAT (Veličković et al., 2018) and then describe our self-supervised GAT (SuperGAT) models.

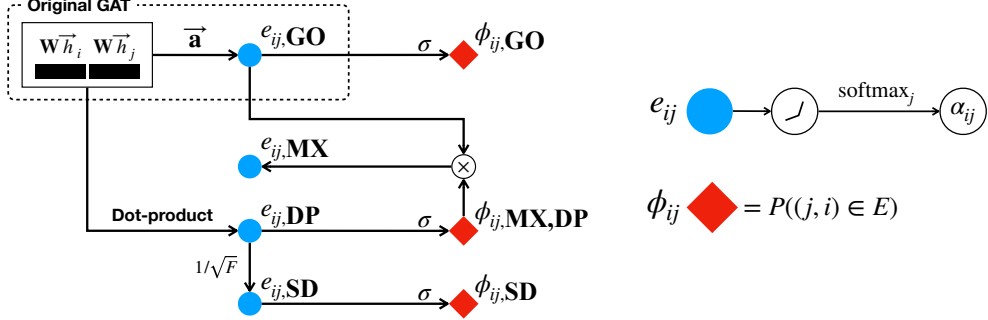

Figure 1: Overview of attention mechanism of SuperGATs: GO, DP, MX, and SD. Blue circles ($e_{ij}$) represent the unnormalized attention before softmax and red diamonds ($\phi_{ij}$) indicate the probability of edge between node $i$ and $j$. The attention mechanism of the original GAT (Veličković et al., 2018) is in the dashed rectangle.

**Notation** For a graph $\mathcal{G} = (V, E)$, $N$ is the number of nodes and $F^l$ is the number of features at layer $l$. Graph attention layer takes a set of features $\mathbb{H}^l = \{\boldsymbol{h}_1^l, \ldots, \boldsymbol{h}_N^l\}, \boldsymbol{h}_i^l \in \mathbb{R}^{F^l}$ as input and produces output features $\mathbb{H}^{l+1} = \{\boldsymbol{h}_1^{l+1}, \ldots, \boldsymbol{h}_N^{l+1}\}$. To compute $\boldsymbol{h}_i^{l+1}$, the model multiplies the weight matrix $\boldsymbol{W}^{l+1} \in \mathbb{R}^{F^{l+1} \times F^l}$ to $\mathbb{H}^l$, linearly combines the features of its first-order neighbors (including itself) $j \in \mathbb{N}_i \cup \{i\}$ by attention coefficients $\alpha_{ij}^{l+1}$, and finally applies a non-linear activation $\rho$. That is $\boldsymbol{h}_i^{l+1} = \rho\left(\sum_{j \in \mathbb{N}_i \cup \{i\}} \alpha_{ij}^{l+1} \boldsymbol{W}^{l+1} \boldsymbol{h}_j^l\right)$. We can compute $\alpha_{ij}^{l+1} = \text{softmax}_j(\text{LReLU}(e_{ij}^{l+1}))$ by normalizing $e_{ij}^{l+1} = a_e(\boldsymbol{W}^{l+1}\boldsymbol{h}_i^l, \boldsymbol{W}^{l+1}\boldsymbol{h}_j^l)$ with softmax on $\mathbb{N}_i \cup \{i\}$ under leaky ReLU activation (Maas et al., 2013), where $a_e$ is a function of the form $\mathbb{R}^{F^{l+1}} \times \mathbb{R}^{F^{l+1}} \to \mathbb{R}$.

**Graph Attention Forms** Among two widely used attention mechanisms, the original GAT (GO) computes the coefficients by single-layer feed-forward network parameterized by $\boldsymbol{a}^{l+1} \in \mathbb{R}^{2F^{l+1}}$. The other is the dot-product (DP) attention, (Luong et al., 2015; Vaswani et al., 2017) motivated by prior work on node representation learning, and it adopts the same mathematical expression for link prediction score (Tang et al., 2015; Kipf & Welling, 2016),

$$e_{ij,\text{GO}}^{l+1} = (\boldsymbol{a}^{l+1})^\top \left[\boldsymbol{W}^{l+1}\boldsymbol{h}_i^l \| \boldsymbol{W}^{l+1}\boldsymbol{h}_j^l\right] \quad \text{and} \quad e_{ij,\text{DP}}^{l+1} = (\boldsymbol{W}^{l+1}\boldsymbol{h}_i^l)^\top \cdot \boldsymbol{W}^{l+1}\boldsymbol{h}_j^l. \quad (1)$$

From now on, we call GAT that uses GO and DP as GAT$_{\text{GO}}$ and GAT$_{\text{DP}}$, respectively.

**Self-supervised Graph Attention Network** We propose SuperGAT with the idea of guiding attention with the presence or absence of an edge between a node pair. We exploit the link prediction task to self-supervise attention with labels from edges: for a pair $i$ and $j$, 1 if an edge exists and 0 otherwise. We introduce $a_\phi$ with sigmoid $\sigma$ to infer the probability $\phi_{ij}$ of an edge between $i$ and $j$.

$$a_\phi : \mathbb{R}^F \times \mathbb{R}^F \to \mathbb{R} \quad \text{and} \quad \phi_{ij} = P\left((j, i) \in E\right) = \sigma(a_\phi(\boldsymbol{W}\boldsymbol{h}_i, \boldsymbol{W}\boldsymbol{h}_j)) \quad (2)$$

We employ four types (GO, DP, SD, and MX) of SuperGAT based on GO and DP attention. For $a_\phi$, the form of which is the same as $a_e$ in GAT$_{\text{GO}}$ and GAT$_{\text{DP}}$, we name them SuperGAT$_{\text{GO}}$ and SuperGAT$_{\text{DP}}$ respectively. For more advanced versions, we describe SuperGAT$_{\text{SD}}$ (Scaled Dot-product) and SuperGAT$_{\text{MX}}$ (Mixed GO and DP) by unnormalized attention $e_{ij}$ and probability $\phi_{ij}$ that an edge exist between $i$ and $j$.

$$\text{SuperGAT}_{\text{SD}}: e_{ij,\text{SD}} = e_{ij,\text{DP}}/\sqrt{F}, \quad \phi_{ij,\text{SD}} = \sigma(e_{ij,\text{SD}}). \quad (3)$$

$$\text{SuperGAT}_{\text{MX}}: e_{ij,\text{MX}} = e_{ij,\text{GO}} \cdot \sigma(e_{ij,\text{DP}}), \quad \phi_{ij,\text{MX}} = \sigma(e_{ij,\text{DP}}). \quad (4)$$

SuperGAT$_{\text{SD}}$ divides the dot-product of nodes by a square root of dimension as Transformer (Vaswani et al., 2017). This prevents some large values to dominate the entire attention after softmax. SuperGAT$_{\text{MX}}$ multiplies GO and DP attention with sigmoid. The motivation of this form comes from the gating mechanism of Gated Recurrent Units (Cho et al., 2014). Since DP attention with the sigmoid represents the probability of an edge, it can softly drop neighbors that are not likely linked while implicitly assigning importance to the remaining nodes.

Training samples are a set of edges $E$ and the complementary set $E^c = (V \times V) \setminus E$. However, if the number of nodes is large, it is not efficient to use all possible negative cases in $E^c$. So, we use negative sampling as in training word or graph embeddings (Mikolov et al., 2013; Tang et al., 2015; Grover & Leskovec, 2016), arbitrarily choosing a total of $p_n \cdot |E|$ negative samples $E^-$ from $E^c$ where the negative sampling ratio $p_n \in \mathbb{R}^+$ is a hyperparameter. SuperGAT is capable of modeling graphs that are sparse with a sufficiently large number of negative samples (i.e., $|V \times V| \gg |E|$), but this is generally not a problem because most real-world graphs are sparse (Chung, 2010).

We define the optimization objective of layer $l$ as a binary cross-entropy loss $\mathcal{L}_E^l$,

$$\mathcal{L}_E^l = -\tfrac{1}{|E \cup E^-|} \sum_{(j,i) \in E \cup E^-} \mathbf{1}_{(j,i)=1} \cdot \log \phi_{ij}^l + \mathbf{1}_{(j,i)=0} \cdot \log \left(1 - \phi_{ij}^l\right), \tag{5}$$

where $\mathbf{1}$. is an indicator function. We use a subset of $E \cup E^-$ sampled by probability $p_e \in (0, 1]$ (also a hyperparameter) at each training iteration for a regularization effect from randomness.

Finally, we combine cross-entropy loss on node labels ($\mathcal{L}_V$), self-supervised graph attention losses for all $L$ layers ($\mathcal{L}_E^l$), and L2 regularization loss, with mixing coefficients $\lambda_E$ and $\lambda_2$.

$$\mathcal{L} = \mathcal{L}_V + \lambda_E \cdot \sum_{l=1}^L \mathcal{L}_E^l + \lambda_2 \cdot \|\boldsymbol{W}\|_2. \tag{6}$$

We use the same form of multi-head attention in GAT and take the mean of each head's attention value before the sigmoid to compute $\phi_{ij}$. Note that SuperGAT has equivalent time and space complexity as GAT. To compute $\mathcal{L}_E^l$ for one head, we need additional operations in terms of $O(F^l \cdot |E \cup E^-|)$, and we do not need extra parameters.

## 4 EXPERIMENTS

Our primary research objective is to design graph attentions that are effective with edge self-supervision. To do this, we pose four specific research questions. We first analyze what basic graph attentions (GO and DP) learn (RQ1 and 2) and how that can be improved with edge self-supervision (RQ3 and 4). We describe each research question and the corresponding experiment design below.

**RQ1. Does graph attention learn label-agreement?**  First, we evaluate what the graph attentions of $\text{GAT}_{\text{GO}}$ and $\text{GAT}_{\text{DP}}$ learn without edge supervision. For this, we present ground-truth of relational importance and a metric to assess graph attention with ground-truth. Wang et al. (2019a) showed that node representations in the connected component converge to the same value in deep GATs. If there is an edge between nodes with different labels, then it will be hard to distinguish the two corresponding labels with GAT of sufficiently many layers; that is, ideal attention should give all weights to label-agreed neighbors. In that sense, we choose label-agreement between nodes as ground-truth of importance.

We compare label-agreement and graph attention based on Kullback–Leibler divergence of the normalized attention $\boldsymbol{\alpha}_k = [\alpha_{kk}, \alpha_{k1}, \ldots, \alpha_{kJ}]$ with label agreement distribution for the center node $k$ and its neighbors 1 to $J$. The label agreement distribution, $\boldsymbol{\ell}_k = [\ell_{kk}, \ell_{k1}, \ldots, \ell_{kJ}]$ is defined by,

$$\ell_{kj} = \hat{\ell}_{kj} / \sum_s \hat{\ell}_{ks}, \quad \hat{\ell}_{kj} = 1 \text{ (if } k \text{ and } j \text{ have the same label) or } 0 \text{ (otherwise).} \tag{7}$$

We employ KL divergence in Eq. 8, whose value becomes small when attention captures well the label-agreement between a node and its neighbors.

$$\text{KLD}(\boldsymbol{\alpha}_k, \boldsymbol{\ell}_k) = \sum_{j \in \mathbb{N}_k \cup \{k\}} \alpha_{kj} \log(\alpha_{kj} / \ell_{kj}) \tag{8}$$

**RQ2. Is graph attention predictive of edge presence?**  To evaluate how well edge information is encoded in SuperGAT, we conduct link prediction experiments with $\text{SuperGAT}_{\text{GO}}$ and $\text{SuperGAT}_{\text{DP}}$ using $\phi_{ij}$ of the last layer as a predictor. We measure the performance by AUC over multiple runs. Since link prediction performance depends on the mixing coefficient $\lambda_E$ in Eq. 6, we adopt multiple $\lambda_E \in \{10^{-3}, 10^{-2}, \ldots, 10^3\}$. We train with an incomplete set of edges, and test with the missing edges and the same number of negative samples. At the same time, node classification performance is measured with the same settings to see how learning edge presence affects node classification.

**RQ3. Which graph attention should we use for given graphs?**   The above two research questions explore what different graph attention learns with or without supervision of edge presence. Then, which graph attention is effective among them for given graphs? We hypothesize that different graph attention will have different abilities to model graphs under various homophily and average degree. We choose these two properties among various graph statistics because they determine the quality and quantity of labels in our self-supervised task. From the perspective of supervised learning of graph attention with edge labels, the learning result depends on *how noisy labels are* (i.e., how low the homophily is) and *how many labels exist* (i.e., how high the average degree is). So, we generate 144 synthetic graphs (Section 4.1) controlling 9 homophily ($0.1 - 0.9$) and 16 average degree ($1 - 100$) and perform the node classification task in the transductive setting with GCN, $GAT_{GO}$, $SuperGAT_{SD}$, and $SuperGAT_{MX}$.

In RQ3, there are also practical reasons to use the average degree and homophily, out of many graph properties (e.g., diameter, degree sequence, degree distribution, average clustering coefficient). First, the graph property can be computed efficiently even for large graphs. Second, there should be an algorithm that can generate graphs by controlling the property of interest only. Third, the property should be a scalar value because if the synthetic graph space is too wide, it would be impossible to conduct an experiment with sufficient coverage. Average degree and homophily satisfy the above conditions and are suitable for our experiment, unlike some of the other graph properties.

**RQ4. Does design choice based on RQ3 generalize to real-world datasets?**   Experiments on synthetic datasets provide an understanding of graph attention models' performance, but they are oversimplified versions of real-world graphs. Can design choice from synthetic datasets be generalized to real-world datasets, considering more complex structures and rich features in real-world graphs? To answer this question, we conduct experiments on 17 real-world datasets with the various average degree ($1.8 - 35.8$) and homophily ($0.16 - 0.91$), and compare them with synthetic graph experiments in RQ3.

## 4.1   DATASETS

**Real-world datasets**   We use a total of 17 real-world datasets (Cora, CiteSeer, PubMed, Cora-ML, Cora-Full, DBLP, ogbn-arxiv, CS, Physics, Photo, Computers, Wiki-CS, Four-Univ, Chameleon, Crocodile, Flickr, and PPI) in diverse domains (citation, co-authorship, co-purchase, web page, and biology) and scales (2k - 169k nodes). We try to use their original settings as much as possible. To verify research questions 1 and 2, we choose four classic benchmarks: Cora, CiteSeer, PubMed in the transductive setting, and PPI in the inductive setting. See appendix A.1 for detailed description, splits, statistics (including degree and homophily), and references.

**Synthetic datasets**   We generate random partition graphs of $n$ nodes per class and $c$ classes (Fortunato, 2010), using `NetworkX` library (Hagberg et al., 2008). A random partition graph is a graph of communities controlled by two probabilities $p_{in}$ and $p_{out}$. If the nodes have the same class labels, they are connected with $p_{in}$, and otherwise, they are connected with $p_{out}$. To generate a graph with an average degree of $d_{avg} = n \cdot \delta$, we choose $p_{in}$ and $p_{out}$ by $p_{in} + (c-1) \cdot p_{out} = \delta$. The input features of nodes are sampled from overlapping multi-Gaussian distributions (Abu-El-Haija et al., 2019). We set $n$ to 500, $c$ to 10, and choose $d_{avg}$ between 1 and 100, $p_{in}$ from $\{0.1\delta, 0.2\delta, \ldots, 0.9\delta\}$. We use 20 samples per class for training, 500 for validation and 1000 for test.

## 4.2   EXPERIMENTAL SET-UP

We follow the experimental set-up of GAT with minor adjustments. All parameters are initialized by Glorot initialization (Glorot & Bengio, 2010) and optimized by Adam (Kingma & Ba, 2014). We apply L2 regularization, dropout (Srivastava et al., 2014) to features and attention coefficients, and early stopping on validation loss and accuracy. We use ELU (Clevert et al., 2016) as a non-linear activation $\rho$. Unless specified, we employ a two-layer SuperGAT with $F = 8$ features and $K = 8$ attention heads (total 64 features). All models are implemented in PyTorch (Paszke et al., 2019) and PyTorch Geometric (Fey & Lenssen, 2019). See appendix A.5 for detailed model and hyperparameter configurations.

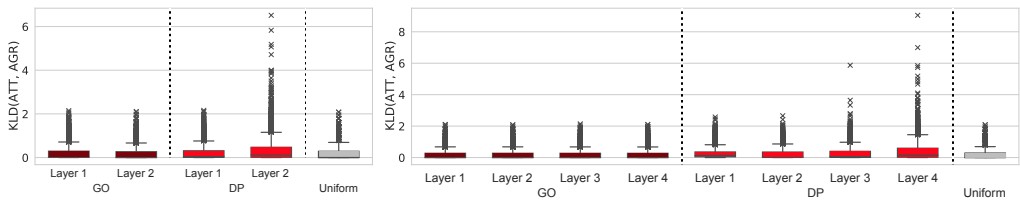

Figure 2: Distribution of KL divergence between normalized attention and label-agreement on all nodes and layers for Cora dataset (Left: two-layer GAT, Right: four-layer GAT).

**Baselines** For all datasets, we compare our model against representative graph neural models: graph convolutional network (GCN) (Kipf & Welling, 2017), GraphSAGE (Hamilton et al., 2017), and graph attention network (GAT) (Veličković et al., 2018). Furthermore, for Cora, CiteSeer, and PubMed, we choose recent graph neural architectures that learn aggregation coefficients (or discrete structures) over edges: constrained graph attention network (CGAT[1]) (Wang et al., 2019a), graph learning-convolutional network (GLCN) (Jiang et al., 2019), learning discrete structure (LDS) (Franceschi et al., 2019), graph agreement model (GAM) (Stretcu et al., 2019), and NeuralSparse (NS in short) (Zheng et al., 2020). For the PPI, we use CGAT as an additional baseline.

## 5 RESULTS

This section describes the experimental results which answer the research questions in Section 4. We include a qualitative analysis of attention, quantitative comparisons on node classification and link prediction, and recipes of graph attention design. The results for sensitivity analysis of important hyper-parameters are in the appendix B.5.

**Does graph attention learn label-agreement?** *GO learns label-agreement better than DP.*

We draw box plots of KL divergence between attention and label agreement distributions of two-layer and four-layer GAT with GO and DP attention for Cora dataset in Figure 2. We see similar patterns in other datasets and place their plots in the appendix B.3. At the rightmost of each sub-figure, we draw the KLD distribution when uniform attention is given to all neighborhoods. Note that the maximum value of KLD of each node is different since the degree of nodes is different. Also, the KLD distribution shows a long-tail shape like a degree distribution of real-world graphs.

There are three observations regarding distributions of KLD. First, we observe that the KLD distribution of GO attention shows a pattern similar to the uniform attention for all citation datasets. This implies that trained GO attention is similar to the uniform distribution, which is in line with previously reported results in the case of entropy[2] (Wang et al., 2019b). Second, KLD values of DP attention tend to be larger than those of GO attention for the last layer, resulting in bigger long-tails. This mismatch between the learned distribution of DP attention and the label agreement distribution suggests that DP attention does not learn label-agreement in the neighborhood. Third, the deeper the model (more than two), the larger the KLD value of DP attention in the last layer. This is because the variance of DP attention increases as the layer gets deeper, as explained below in Proposition 1.

**Proposition 1.** *For $l + 1$th GAT layer, if $\mathbf{W}$ and $\mathbf{a}$ are independent and identically drawn from zero-mean uniform distribution with variance $\sigma_w^2$ and $\sigma_a^2$ respectively, assuming that parameters are independent to input features $\mathbf{h}^l$ and elements of $\mathbf{h}^l$ are independent to each other,*

$$Var[e_{ij,GO}^{l+1}] = 2F^{l+1}\sigma_w^2\sigma_a^2\mathbb{E}(\|\mathbf{h}^l\|_2^2) \ and \ Var[e_{ij,DP}^{l+1}] \geq F^{l+1}\sigma_w^4\left(\tfrac{4}{5}\mathbb{E}\left(((\mathbf{h}_i^l)^\top\mathbf{h}_j^l)^2\right) + Var((\mathbf{h}_i^l)^\top\mathbf{h}_j^l)\right) \quad (9)$$

The proof is given in the appendix B.1. While the variance of GO depends on the norm of features only, the variance of DP depends on the expectation of the square of input's dot-product and variance of input's dot-product. Stacking GAT layers, the more features of $i$ and $j$ correlate with each other, the larger the input's dot-product will be. After DP attention is normalized by softmax, which intensifies the larger values among them, normalized DP attention attends to only a small portion of the neighbors and learns a biased representation.

---

[1]Since CGAT uses node labels in the loss function, it is difficult to use it in semi-supervised learning. So, we modify its auxiliary loss for SSL. See appendix A.6 for details.

[2]https://docs.dgl.ai/en/latest/tutorials/models/1_gnn/9_gat.html

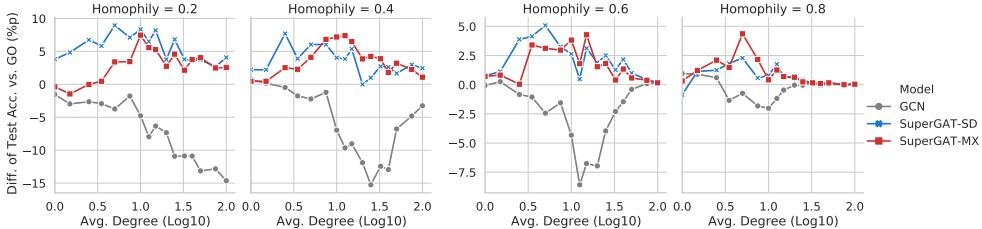

Figure 3: Test performance on node classification and link prediction for GO and DP attentions against the mixing coefficient $\lambda_E$. We report accuracy (Cora, CiteSeer, PubMed) and micro f1-score (PPI) for node classification, and AUC for link prediction.

Figure 4: Mean test accuracy gains (of 5 runs) against $\text{GAT}_{\text{GO}}$ on synthetic datasets, varying homophily and average degree of the input graph.

**Is graph attention predictive for edge presence?**    *DP predicts edge presence better than GO.*

In Figure 3, we report the mean AUC over multiple runs (5 for PPI and 10 for others) for link prediction (red lines) and node classification (gray lines). As the mixing coefficient $\lambda_E$ increases, the link prediction score increases in all datasets and attentions. This is a natural result considering that $\lambda_E$ is the weight factor of self-supervised graph attention loss ($\mathcal{L}_E$ in Equation 6). For three out of four datasets, DP attention outperforms GO for link prediction for all range of $\lambda_E$ in our experiment. Surprisingly, even for small $\lambda_E$, DP attention shows around 80 AUC, much higher than the performance of GO attention. PPI is an exception where GO attention shows higher performance for small $\lambda_E$ than DP, but the difference is slight. The results of this experiment demonstrate that DP attention is more suitable than GO attention in encoding edges.

This figure also includes node classification performance. For all datasets except PubMed, we observe a trade-off between node classification and link prediction; that is, node classification performance decreases in $\text{SuperGAT}_{\text{GO}}$ and $\text{SuperGAT}_{\text{DP}}$ as $\lambda_E$ increases and thus link prediction performance increases. PubMed also shows a decrease in performance at the largest $\lambda_E$ we have tested. This implies that it is hard to learn the relational importance from edges by simply optimizing graph attention for link prediction.

**Which graph attention should we use for given graphs?**    *It depends on homophily and average degree of the graph.*

In Figure 4, we draw the mean test accuracy gains (over 5 runs) against $\text{GAT}_{\text{GO}}$ as the average degree increases from 1 to 100, for different values of homophily, on 64 synthetic graphs with GCN, $\text{SuperGAT}_{\text{SD}}$, $\text{SuperGAT}_{\text{MX}}$. See full results from appendix B.2. We define homophily $h$ as the average ratio of neighbors with the same label as the center node (Pei et al., 2020). That is $h = \frac{1}{|V|} \sum_{i \in V} \left( \sum_{j \in \mathbb{N}_i} \mathbf{1}_{l(i)=l(j)} / |\mathbb{N}_i| \right)$, where $l(i)$ is the label of node $i$. The expectation of homophily for random partition graphs is analytically $p_{in}/\delta$, and we just adopt this value to label the homophily of graphs in Figure 4.

We make the following observations from this figure. First, if the homophily is low ($\leq 0.2$), $\text{SuperGAT}_{\text{SD}}$ performs best among models because DP attention tends to focus on a small number of neighbors. This result empirically confirms what we analytically found in Proposition 1. Second, even when homophily is low, the performance gain of SuperGAT against GAT increases as the average degree increases to a certain level (around 10), meaning relation modeling can benefit from self-supervision if there are sufficiently many edges providing supervision. This is in agreement with prior study of label noise for deep neural networks where they find that the absolute amount of data with correct labels affects the learning quality more than the ratio between data with noisy and correct labels (Rolnick et al., 2017). Third, if the average degree and homophily are high enough,

Table 1: Summary of classification accuracies of GCN, GraphSAGE, GAT, SuperGAT$_{SD}$, and SuperGAT$_{MX}$ for real-world datasets (30 runs for ogbn-arxiv and Flickr, and 100 runs for others).

| Model | ogbn-arxiv | CS | Physics | Cora-ML | Cora-Full | DBLP | Cham. | Four-Univ | Wiki-CS | Photo | Comp. | Flickr | Croco. |
|---|---|---|---|---|---|---|---|---|---|---|---|---|---|
| GCN | $33.3_{\pm 1.2}$ | $91.5_{\pm 0.2}$ | $92.5_{\pm 0.2}$ | $85.0_{\pm 0.4}$ | $59.5_{\pm 0.2}$ | $77.8_{\pm 0.5}$ | $33.1_{\pm 0.9}$ | $74.8_{\pm 0.6}$ | $74.0_{\pm 1.0}$ | $91.6_{\pm 0.6}$ | $84.5_{\pm 1.4}$ | $51.2_{\pm 0.4}$ | $32.6_{\pm 0.4}$ |
| GraphSAGE | $54.6_{\pm 0.3}$ | $90.0_{\pm 0.1}$ | $92.2_{\pm 0.1}$ | $83.7_{\pm 0.4}$ | $59.2_{\pm 0.2}$ | $78.7_{\pm 0.6}$ | $41.0_{\pm 0.9}$ | $74.4_{\pm 0.6}$ | $77.5_{\pm 0.5}$ | $90.4_{\pm 1.1}$ | $83.0_{\pm 1.4}$ | $50.7_{\pm 0.2}$ | $53.0_{\pm 1.0}$ |
| GAT | $54.1_{\pm 0.5}$ | $89.5_{\pm 0.2}$ | $91.2_{\pm 0.6}$ | $83.2_{\pm 0.6}$ | $58.7_{\pm 0.3}$ | $78.2_{\pm 1.5}$ | $40.8_{\pm 0.7}$ | $74.2_{\pm 0.7}$ | $77.6_{\pm 0.6}$ | $\mathbf{91.8}_{\pm 0.6}$ | $\mathbf{85.7}_{\pm 0.9}$ | $\mathbf{50.9}_{\pm 0.4}$ | $\mathbf{53.3}_{\pm 1.0}$ |
| SuperGAT$_{SD}$ | $54.5^{**}_{\pm 0.3}$ | $88.8^{\downarrow}_{\pm 0.2}$ | $91.6^{**}_{\pm 0.5}$ | $84.5^{**}_{\pm 0.4}$ | $55.8^{\downarrow}_{\pm 0.6}$ | $79.4^{**}_{\pm 0.8}$ | $41.6^{**}_{\pm 0.7}$ | $\mathbf{76.2}^{**}_{\pm 0.8}$ | $77.9_{\pm 0.7}$ | $86.8^{\downarrow}_{\pm 2.5}$ | $82.2^{\downarrow}_{\pm 0.9}$ | $45.1^{\downarrow}_{\pm 1.2}$ | $\mathbf{53.3}_{\pm 0.9}$ |
| SuperGAT$_{MX}$ | $\mathbf{55.1}^{**}_{\pm 0.2}$ | $\mathbf{90.2}^{**}_{\pm 0.2}$ | $\mathbf{91.9}^{**}_{\pm 0.5}$ | $\mathbf{84.7}^{**}_{\pm 0.4}$ | $\mathbf{59.6}^{**}_{\pm 0.2}$ | $\mathbf{80.7}^{**}_{\pm 0.7}$ | $\mathbf{42.0}^{**}_{\pm 0.8}$ | $75.3^{\downarrow}_{\pm 0.6}$ | $77.9_{\pm 0.5}$ | $\mathbf{91.8}_{\pm 0.9}$ | $\mathbf{85.7}_{\pm 1.1}$ | $50.8^{\downarrow}_{\pm 0.2}$ | $\mathbf{53.3}_{\pm 0.9}$ |

Table 2: Summary of classification accuracies with 100 random seeds for Cora, CiteSeer, and PubMed. We mark with daggers (†) the reprinted results from the respective papers.

| Model | Cora | CiteSeer | PubMed |
|---|---|---|---|
| GCN† | 81.5 | 70.3 | 79.0 |
| GraphSAGE | $82.1 \pm 0.6$ | $71.9 \pm 0.9$ | $78.0 \pm 0.7$ |
| CGAT | $81.4 \pm 1.1$ | $70.1 \pm 0.9$ | $78.1 \pm 1.0$ |
| GLCN† | 85.5 | 72.0 | 78.3 |
| LDS† | 84.1 | 75 | - |
| GCN + GAM† | 86.2 | 73.5 | 86.0 |
| GCN + NS† | $83.7 \pm 1.4$ | $74.1 \pm 1.4$ | - |
| GAT† | $83.0 \pm 0.7$ | $72.5 \pm 0.7$ | $79.0 \pm 0.4$ |
| SuperGAT$_{SD}$ | $82.7^{\downarrow}_{\pm 0.6}$ | $72.5_{\pm 0.8}$ | $81.3^{**}_{\pm 0.5}$ |
| SuperGAT$_{MX}$ | $\mathbf{84.3}^{**}_{\pm 0.6}$ | $\mathbf{72.6}_{\pm 0.8}$ | $\mathbf{81.7}^{**}_{\pm 0.5}$ |

Table 3: Summary of micro f1-scores with 30 random seeds for PPI.

| Model | PPI |
|---|---|
| GCN | $61.5 \pm 0.4$ |
| GraphSAGE | $59.0 \pm 1.2$ |
| CGAT | $68.3 \pm 1.7$ |
| GAT | $72.2 \pm 0.6$ |
| SuperGAT$_{SD}$ | $\mathbf{74.4}^{**}_{\pm 0.4}$ |
| SuperGAT$_{MX}$ | $67.2^{\downarrow}_{\pm 1.2}$ |

| | |
|---|---|
| ** | $p$-value $< .0001$ |
| * | $p$-value $< .0005$ |
| ↓ | Worse than GAT$_{GO}$ |
| Color | Best graph attention (See Fig. 5) |

there is no difference between all models, including GCNs. If there are more correct edges beyond a certain amount, we can learn fine representation without self-supervision. Most importantly, if the average degree is not too low or high and homophily is above $0.2$, SuperGAT$_{MX}$ performs better than or similar to SuperGAT$_{SD}$. This implies that we can take advantage of both GO attention to learn label-agreement and DP attention to learn edge presence by mixing GO and DP. Note that many of the real-world graphs belong to this range of graph characteristics.

The results on synthetic graphs imply that understanding of graph domains should be preceded to design graph attention. That is, by knowing the average degree and homophily of the graphs, we can choose the optimal graph attention in our design space.

**Does design choice based on RQ3 generalize to real-world datasets?** *It does for 15 of 17 real-world datasets.*

In Figure 5, we plot the best-performed graph attention for synthetic graphs with square points in the plane of average degree and homophily. The size is the performance gain of SuperGAT against GAT, and the color indicates the best model. If the difference is not statistically significant ($p$-value $\geq .05$) between GAT and SuperGAT, and between SuperGAT$_{MX}$ and SuperGAT$_{SD}$, we mark as GAT-Any and SuperGAT-Any, respectively. We call this plot a recipe since it introduces the optimal attention to a specific region's graph.

Now we map the results of 17 real-world datasets in Figure 5 according to their average degree and homophily. Average degree and homophily can be found in the appendix A.1, and experimental results of graph attention models are summarized in Tables 1, 2 and 3. We report the mean and standard deviation of performance over multiple seeds (30 for graphs with more than 50k nodes and 100 for others). We put unpaired t-test results of SuperGAT with GAT$_{GO}$ with asterisks.

We find that the graph attention recipe based on synthetic experiments can generalize across real-world graphs. PPI and Four-Univ (▲) are surrounded by squares of SuperGAT$_{SD}$ (■) at the bottom of the plane. Wiki-CS (▲), located in the SuperGAT-Any region (■), also show no difference in performance between SuperGATs. Nine datasets (▲), which SuperGAT$_{MX}$ shows the highest performance, are located in the MX regions (■) or within the margin of two squares. Note that there are two MX regions: lower-middle average degree (2.5 - 7.5) and high homophily (0.8 - 0.9), and upper-middle average degree (7.5 - 50) and lower-middle homophily (0.3 - 0.5). There are five datasets with no significant performance change across graph attention (▲). CiteSeer, Photo, and Computers are within a margin of one square from the GAT-Any region (■); however, Flickr and Crocodile are

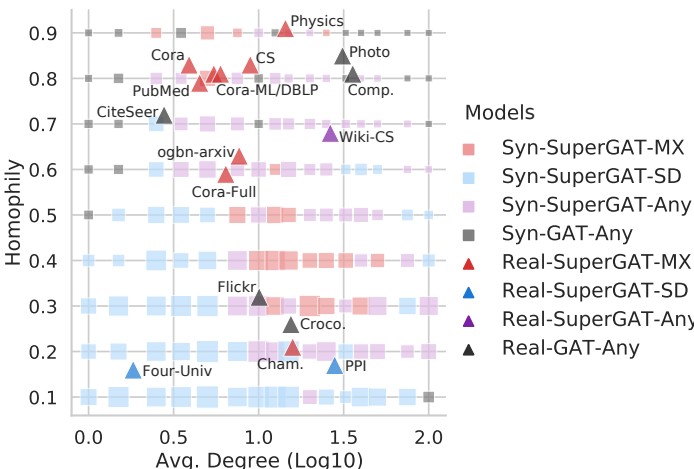

Figure 5: The best-performed graph attention design for synthetic and real-world graphs with various average degree and homophily.

in the SuperGAT-Any region. To find out the cause of this irregularity, we examine the distribution of degree and per-node homophily (appendix A.4). We observe a more complex mixture distribution of homophily and average degree in Flickr and Crocodile, and this seems to be equivalent to mixing graphs of different characteristics, resulting in inconsistent results with our attention design recipe.

**Comparison with baselines**  For a total of 17 datasets, SuperGAT outperforms GCN for 13 datasets, GAT for 12 datasets, GraphSAGE for 16 datasets. Interestingly, for CS, Physics, Cora-ML, and Flickr, in which our model performs worse than GCN, GAT also cannot surpass GCN. It is not yet known when the degree-normalized aggregation of GCN outperforms the attention-based aggregation, and more research is needed to figure out how to embed the degree information into graph attention. Tables 2 and 3 show performance comparisons between SuperGAT and recent GNNs for Cora, CiteSeer, PubMed, and PPI. Our model performs better for CiteSeer (0.6%p) and PubMed (3.4%p) than GLCN, which gives regularization to all relations in a graph. GCN + NS (NeuralSparse) performs better than our model for CiteSeer (1.5%p) but not for Cora (0.6%p). CGAT modified for semi-supervised learning shows lower performance than GAT. Although LDS and GAM which use iterative training show better performance except for LDS on Cora, these models require significantly more computation for the iterative training. For example, GCN + GAM compared to our model needs more ×34 more training time for Cora, ×72 for CiteSeer, and ×82 for PubMed. See appendix A.7 and B.4 for the experimental set-up and the result of wall-clock time analysis.

## 6 CONCLUSION

We proposed novel graph neural architecture designs to self-supervise graph attention following the input graph's characteristics. We first assessed what graph attention is learning and analyzed the effect of edge self-supervision to link prediction and node classification performance. This analysis showed two widely used attention mechanisms (original GAT and dot-product) have difficulty encoding label-agreement and edge presence simultaneously. To address this problem, we suggested several graph attention forms that balance these two factors and argued that graph attention should be designed depending on the input graph's average degree and homophily. Our experiments demonstrated that our graph attention recipe generalizes across various real-world datasets such that the models designed according to the recipe outperform other baseline models.

### ACKNOWLEDGMENTS

This research was supported by the Engineering Research Center Program through the National Research Foundation of Korea (NRF) funded by the Korean Government MSIT (NRF-2018R1A5A1059921)

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

## A  EXPERIMENTAL SET-UP

### A.1  REAL-WORLD DATASET

In this section, we describe details (including nodes, edges, features, labels, and splits) of real-world datasets. We report statistics of real-world datasets in Tables 4 and 5. For multi-label graphs (PPI), we extend the homophily to the average of the ratio of shared labels on neighbors over nodes, i.e., $h = \frac{1}{|V|} \sum_{i \in V} \left( \sum_{j \in \mathbb{N}_i} |\mathbb{C}_i \cap \mathbb{C}_j| / (|\mathbb{N}_i| \cdot |\mathbb{C}|) \right)$, where $\mathbb{C}_i$ is a set of labels for node $i$ and $\mathbb{C}$ is a set of all labels.

#### A.1.1  CITATION NETWORK

We use a total of 7 citation network datasets. Nodes are documents, and edges are citations. The task for all citation network datasets is to classify each paper's topic.

**Cora, CiteSeer, PubMed**    We use three benchmark datasets for semi-supervised node classification tasks in the transductive setting (Sen et al., 2008; Yang et al., 2016). The features of the nodes are bag-of-words representations of documents. We follow the train/validation/test split of previous work (Kipf & Welling, 2017). We use 20 samples per class for training, 500 samples for the validation, and 1000 samples for the test.

**Cora-ML, Cora-Full, DBLP**    These are other citation network datasets from Bojchevski & Günnemann (2018). Node features are bag-of-words representations of documents. For CoraFull with features more than 5000, we reduce the dimension to 500 by performing PCA. We use the split setting in Shchur et al. (2018): 20 samples per class for training, 30 samples per class for validation, the rest for the test.

**ogbn-arxiv**    The ogbn-arxiv is a recently proposed large-scale dataset of citation networks (Hu et al., 2020a; Wang et al., 2020). Nodes represent arXiv papers, and edges indicate citations between papers, and node features are mean vectors of skip-gram word embeddings of their titles and abstracts. We use the public split by publication dates provided by the original paper.

#### A.1.2  CO-AUTHOR NETWORK

**CS, Physics**    The CS and Physics are co-author networks in each domain (Shchur et al., 2018). Nodes are authors, and edges mean whether two authors co-authored a paper. Node features are paper keywords from the author's papers, and we reduce the original dimension (6805 and 8415) to 500 using PCA. The split is the 20-per-class/30-per-class/rest from Shchur et al. (2018). The goal of this task is to classify each author's respective field of study.

#### A.1.3  AMAZON CO-PURCHASE

**Photo, Computers**    The Photo and Computers are parts of the Amazon co-purchase graph (McAuley et al., 2015; Shchur et al., 2018). Nodes are goods, and edges indicate whether two goods are frequently purchased together, and node features are a bag-of-words representation of product reviews. The split is the 20-per-class/30-per-class/rest from Shchur et al. (2018). The task is to classify the categories of goods.

#### A.1.4  WEB PAGE NETWORK

**Wiki-CS**    The Wiki-CS dataset is computer science related page networks in Wikipedia (Mernyei & Cangea, 2020). Nodes represent articles about computer science, and edges represent hyperlinks between articles. The features of nodes are mean vectors of GloVe word embeddings of articles. There are 20 standard splits, and we experiment with five random seeds for each split (total 100 runs). The task is to classify the main category of articles.

**Chameleon, Crocodile**    These datasets are Wikipedia page networks about specific topics, Chameleon and Crocodile (Rozemberczki et al., 2019). Nodes are articles, and edges are mutual

Table 4: Average degree and homophily of real-world graphs.

| Dataset | Degree | Homophily |
|---------|--------|-----------|
| Four-Univ | $1.83 \pm 1.71$ | 0.16 |
| PPI | $28.0 \pm 39.26$ | 0.17 |
| Chameleon | $15.85 \pm 18.20$ | 0.21 |
| Crocodile | $15.48 \pm 15.97$ | 0.26 |
| Flickr | $10.08 \pm 31.75$ | 0.32 |
| Cora-Full | $6.41 \pm 8.79$ | 0.59 |
| ogbn-arxiv | $7.68 \pm 9.05$ | 0.63 |
| Wiki-CS | $26.40 \pm 36.04$ | 0.68 |
| CiteSeer | $2.78 \pm 3.39$ | 0.72 |
| PubMed | $4.50 \pm 7.43$ | 0.79 |
| Cora-ML | $5.45 \pm 8.24$ | 0.81 |
| DBLP | $5.97 \pm 9.35$ | 0.81 |
| Computers | $35.76 \pm 70.31$ | 0.81 |
| Cora | $3.90 \pm 5.23$ | 0.83 |
| CS | $8.93 \pm 9.11$ | 0.83 |
| Photo | $31.13 \pm 47.27$ | 0.85 |
| Physics | $14.38 \pm 15.57$ | 0.91 |

links between them. Node features are a bag-of-words representation with informative nouns in the article. The number of features is 13183, but we use a reduced dimension of 500 by PCA. The split is 20-per-class/30-per-class/rest from Shchur et al. (2018). The original dataset is for the regression task to predict monthly traffic, but we group values into six bins and make it a classification problem.

**Four-Univ** The Four-Univ dataset is a web page networks from computer science departments of diverse universities (Craven et al., 1998). Nodes are web pages, edges are hyperlinks between them, and node features are TF-IDF vectors of web page's contents. There are five graphs consists of four universities (Cornell, Texas, Washington, and Wisconsin) and a miscellaneous graph from other universities. As the original authors suggested[3], we use three graphs of universities and a miscellaneous graph for training, another one graph for validation (Cornell), and the other one graph for the test (Texas). Classification labels are types of web pages (student, faculty, staff, department, course, and project).

### A.1.5 FLICKR

The Flickr dataset is a graph of images from Flickr (McAuley & Leskovec, 2012; Zeng et al., 2020). Nodes are images, and edges indicate whether two images share common properties such as geographic location, gallery, and users commented. Node features are a bag-of-words representation of images. We use labels and split in in Zeng et al. (2020). For labels, they construct seven classes by manually merging 81 image tags.

### A.1.6 PROTEIN-PROTEIN INTERACTION

The protein-protein interaction (PPI) dataset (Zitnik & Leskovec, 2017; Hamilton et al., 2017; Subramanian et al., 2005) is a well-known benchmark in the inductive setting. A graph is given for human tissue, the nodes are proteins, the node's features are biological signatures like genes, and the edges illustrate proteins' interactions. The dataset consists of 20 training graphs, two validation graphs, and two test graphs. This dataset has multi-labels of gene ontology sets.

### A.2 LINK PREDICTION

For link prediction, we split 5% and 10% of edges for validation and test set, respectively. We fix the negative edges for the test set and sample negative edges for the training set at each iteration.

---

[3]http://www.cs.cmu.edu/afs/cs.cmu.edu/project/theo-20/www/data/

Table 5: Statistics of the real-world datasets.

| Dataset | # Nodes | # Edges | # Features | # Classes | Split | # Training Nodes | # Val. Nodes | # Test Nodes |
|---|---|---|---|---|---|---|---|---|
| Four-Univ | 4518 | 3426 | 2000 | 6 | fixed | 4014 (3 Gs) | 248 (1 G) | 256 (1 G) |
| PPI | 56944 | 818716 | 50 | 121 | fixed | 44906 (20 Gs) | 6514 (2 Gs) | 5524 (2 Gs) |
| Chameleon | 2277 | 36101 | 500 | 6 | random | 120 | 180 | 1977 |
| Crocodile | 11631 | 180020 | 500 | 6 | random | 120 | 180 | 11331 |
| Flickr | 89250 | 449878 | 500 | 7 | fixed | 44625 | 22312 | 22313 |
| Cora-Full | 19793 | 63421 | 500 | 70 | random | 1395 | 2049 | 16349 |
| ogbn-arxiv | 169343 | 1166243 | 128 | 40 | fixed | 90941 | 29799 | 48603 |
| Wiki-CS | 11701 | 297110 | 300 | 10 | fixed | 580 | 1769 | 5847 |
| CiteSeer | 3327 | 4732 | 3703 | 6 | fixed | 120 | 500 | 1000 |
| PubMed | 19717 | 44338 | 500 | 3 | fixed | 60 | 500 | 1000 |
| Cora-ML | 2995 | 8158 | 2879 | 7 | random | 140 | 210 | 2645 |
| DBLP | 17716 | 52867 | 1639 | 4 | random | 80 | 120 | 17516 |
| Computers | 13752 | 245861 | 767 | 10 | random | 200 | 300 | 13252 |
| Cora | 2708 | 5429 | 1433 | 7 | fixed | 140 | 500 | 1000 |
| CS | 18333 | 81894 | 500 | 15 | random | 300 | 450 | 17583 |
| Photo | 7650 | 119081 | 745 | 8 | random | 160 | 240 | 7250 |
| Physics | 34493 | 247962 | 500 | 5 | random | 100 | 150 | 34243 |

## A.3 SYNTHETIC DATASET

To the best of our knowledge, our synthetic datasets are not used in recent literature. Therefore, we give some small examples of synthetic datasets to see qualitatively how the average degree and homophily vary according to $\delta$ and $p_{in}$. Specifically, we draw 2D t-SNE plot of node features and edges in Figure 6. In this figure, we can observe that the average degree ($d_{avg} = n \cdot \delta$) increases as $\delta$ increases, and homophily ($h = p_{in}/\delta$) increases as $p_{in}$ increases. Note that these are raw input features sampled from the 2D Gaussian distribution, not learned node representation. We use the code from the prior work[4] (Abu-El-Haija et al., 2019) and apply normalization by standard score. We choose $\delta$ from $\{0.025, 0.2\}$, $p_{in}$ from $\{0.1\delta, 0.5\delta, 0.9\delta\}$, and fix $n = 100$ and $c = 5$.

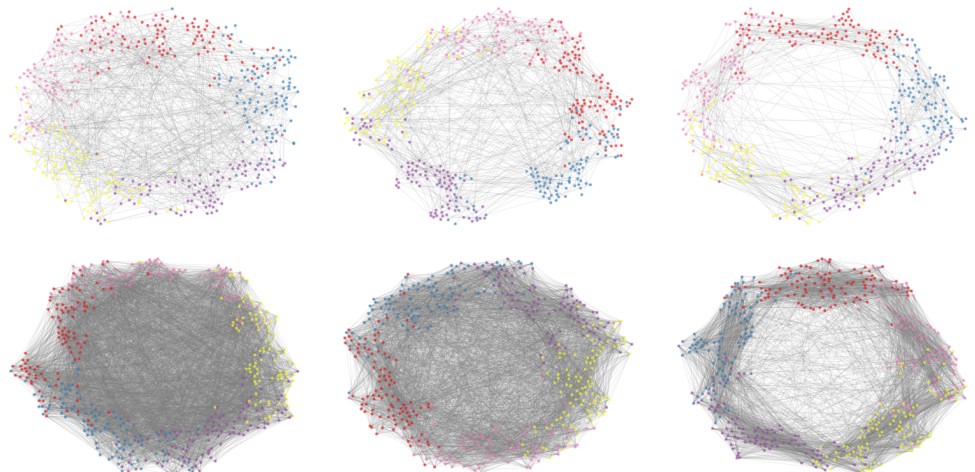

Figure 6: t-SNE plots of node features and edges for synthetic graph examples. Hyperparameters are $\delta \in \{0.025 \text{ (Top)}, 0.2 \text{ (Bottom)}\}$ and $p_{in} \in \{0.1\delta \text{ (Left)}, 0.5\delta \text{ (Center)}, 0.9\delta \text{ (Right)}\}$.

## A.4 DISTRIBUTION OF DEGREE AND HOMOPHILY OF DATASETS

In Figure 7, we draw kernel density estimation plots of per-node homophily and degree of nodes in real-world datasets. We define per-node homophily as the ratio of neighbors with the same label as the center node, that is, $h_i = \sum_{j \in \mathbb{N}_i} \mathbf{1}_{l(i)=l(j)} / |\mathbb{N}_i|$. Note that we define homophily as $h = \frac{1}{|V|} \sum_{i \in V} h_i$ in Section 5.

---

[4] https://github.com/samihaija/mixhop/blob/master/data/synthetic/make_x.py

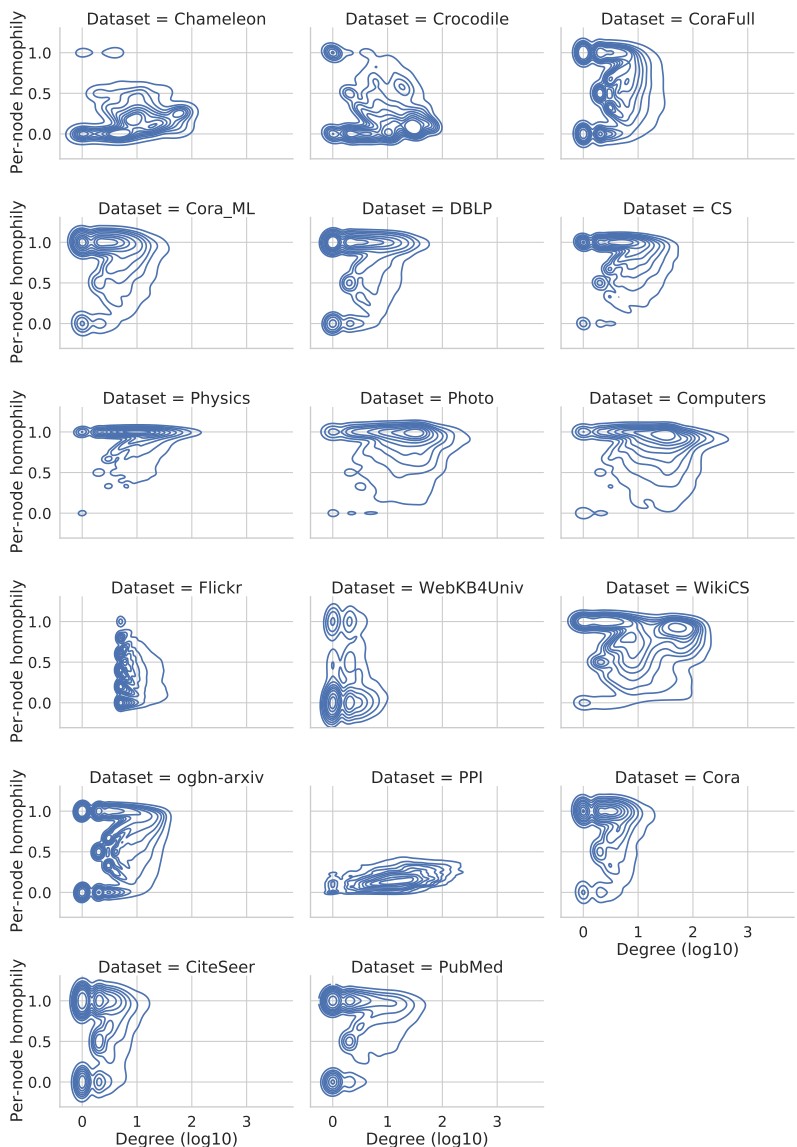

Figure 7: Kernel density estimate plot of distribution of degree and per-node homophily in real-world graphs.

We focus more on the part outside of where the degree is 1 (0 with log scale), and the per-node homophily is 0. These are leaf nodes incorrectly connected and does not significantly affect the learning overall graph representation. In most datasets, only the largest mode exists, or there are some small modes around it. However, in Flickr and Crocodile, we can observe that the interval between modes is wide. More specifically, Crocodile's modes are in the area of (high degree, low per-node homophily) and (low degree, high per-node homophily), and Flickr's modes cover most homophily at a specific degree. Note that we can regard a mixture of distribution as a mixture of different sub-graphs. We argue that this is the reason our recipe does not fit for these two datasets.

## A.5    MODEL & HYPERPARAMETER CONFIGURATIONS

**Model**    Since we experiment with numbers of datasets, we maintain almost the same configurations across datasets. We do not use other methods such as residual connections, deeper layers, batch normalization, edge augmentation, and more hidden features, although we have confirmed from previous studies that these techniques contribute to performance improvement. For example, prior

Table 6: Hyperparameters for experiments on real-world datasets.

| Dataset | Model | $\lambda_2$ | $\lambda_E$ | $p_e$ | $p_n$ |
|---|---|---|---|---|---|
| Cora | SuperGAT$_{SD}$ | 0.007829935945 | 10.88266937 | 0.8 | 0.5 |
| | SuperGAT$_{MX}$ | 0.008228864973 | 11.34657453 | 0.8 | 0.5 |
| CiteSeer | SuperGAT$_{SD}$ | 0.04823808657 | 0.09073992828 | 0.8 | 0.3 |
| | SuperGAT$_{MX}$ | 0.04161321832 | 0.01308169273 | 0.8 | 0.5 |
| PubMed | SuperGAT$_{SD}$ | 0.0002030927563 | 18.82560333 | 0.6 | 0.7 |
| | SuperGAT$_{MX}$ | 2.19E-04 | 10.4520518 | 0.6 | 0.5 |
| PPI | SuperGAT$_{SD}$ | 3.39E-07 | 0.001034351842 | 1 | 0.5 |
| | SuperGAT$_{MX}$ | 1.00E-07 | 1.79E-06 | 1 | 0.5 |

Table 7: Summary of classification accuracies with 10 random seeds for Cora, CiteSeer, and PubMed in the full-supervised setting. We mark with asterisks the reprinted results from the respective papers.

| Model | Cora | CiteSeer | PubMed |
|---|---|---|---|
| GAT* | $87.2 \pm 0.3$ | $77.3 \pm 0.3$ | $87.0 \pm 0.3$ |
| CGAT* | $88.2 \pm 0.3$ | $78.9 \pm 0.2$ | $87.4 \pm 0.3$ |
| CGAT (Our Impl.: 2-layer w/ 64-features) | $88.9 \pm 0.3$ | $78.9 \pm 0.2$ | $86.9 \pm 0.2$ |
| SuperGAT$_{MX}$ (2-layer w/ 64-features) | $88.7 \pm 0.2$ | $79.1 \pm 0.2$ | $87.0 \pm 0.1$ |

work has shown f1-score close to 100 for PPI. To clearly see the difference between the various graph attention designs, we intentionally keep a simple model configuration.

**Hyperparameter** For real-world datasets, we tune two hyperparameters (mixing coefficients $\lambda_2$ and $\lambda_E$) by Bayesian optimization for the mean performance of 3 random seeds. We choose negative sampling ratio $p_n$ from $\{0.3, 0.5, 0.7, 0.9\}$, and edge sampling ratio $p_e$ from $\{0.6, 0.8, 1.0\}$. We fix dropout probability to 0.0 for PPI, 0.2 for ogbn-arxiv, 0.6 for others. We set learning rate to 0.05 (ogbn-arxiv), 0.01 (PubMed, PPI, Wiki-CS, Photo, Computers, CS, Physics, Crocodile, Cora-Full, DBLP), 0.005 (Cora, CiteSeer, Cora-ML, Chameleon), 0.001 (Four-Univ). For ogbn-arxiv, we set the number of features per head to 16 and the number of heads in the last layer to one; otherwise, we use eight features per head and eight heads in the last layer.

For synthetic datasets, we choose $\lambda_E$ from $\{10^{-5}, 10^{-4}, 10^{-3}, 10^{-2}, 10^{-1}, 1, 10, 10^2\}$ and $\lambda_2$ from $\{10^{-7}, 10^{-5}, 10^{-3}\}$. We fix learning rate to 0.01, dropout probability to 0.2, $p_n$ to 0.5, and, $p_e$ to 0.8 for all synthetic graphs.

Table 6 describes hyperparameters for SuperGAT on four real-world datasets. For other datasets and experiments, please see the code (`./SuperGAT/args.yaml`).

## A.6 CGAT IMPLEMENTATION

CGAT (Wang et al., 2019a) has two auxiliary losses: graph structure based constraint $\mathcal{L}_g$ and class boundary constraint $\mathcal{L}_b$. We borrow their notation for this section: $\mathcal{V}$ as a set of nodes, $\mathbb{N}_i$ as a set of one-hop neighbors, $\mathbb{N}_i^+$ as a set of neighbors that share labels, $\mathbb{N}_i^-$ as a set of neighbors that do not share labels, $\zeta.$ as a margin between attention values, and $\phi(v_i, v_k)$ as unnormalized attention value.

$$\mathcal{L}_g = \sum_{i \in \mathcal{V}} \sum_{j \in \mathbb{N}_i \setminus \mathbb{N}_i^-} \sum_{k \in (\mathcal{V} \setminus \mathbb{N}_i)} \max\left(0, \phi(v_i, v_k) + \zeta_g - \phi(v_i, v_j)\right) \tag{10}$$

$$\mathcal{L}_b = \sum_{i \in \mathcal{V}} \sum_{j \in \mathbb{N}_i^+} \sum_{k \in \mathbb{N}_i^-} \max\left(0, \phi(v_i, v_k) + \zeta_b - \phi(v_i, v_j)\right) \tag{11}$$

Since label information is included in these two losses, they are difficult to use in semi-supervised settings that provide few labeled samples. In fact, in the CGAT paper, they conduct experiments in full-supervised settings; that is, they use all nodes in training except validation and test nodes.

So, we only use $\mathcal{L}_g$ modified for semi-supervised learning.

$$\mathcal{L}_g^{\text{SSL}} = \sum_{i \in \mathcal{V}} \sum_{j \in \mathbb{N}_i} \sum_{k \in (\mathcal{V} \backslash \mathbb{N}_i)} \max\left(0, \phi\left(v_i, v_k\right) + \zeta_g - \phi\left(v_i, v_j\right)\right) \quad (12)$$

With $\mathcal{L}_c$, the multi-class cross-entropy on node labels, CGAT's optimization objective is

$$\mathcal{L} = \mathcal{L}_c + \lambda_g \mathcal{L}_g + \lambda_b \mathcal{L}_b, \quad (13)$$

and our modified CGAT's loss is

$$\mathcal{L} = \mathcal{L}_c + \lambda_g \mathcal{L}_g^{\text{SSL}}. \quad (14)$$

In addition to losses, CGAT proposes top-k softmax and node importance based negative sampling (NINS). Top-k softmax picks up nodes with top-k attention values among neighbors. NINS adopts importance sampling when choosing negative sample nodes.

Since the code for CGAT has not been released, we implement our own version. In all experiments in our paper, we use only the modified losses (Equation 14) and top-k softmax due to the training and implementation complexity of NINS. For PPI, even if we do not assume a semi-supervised setting, we use the same loss because we could not accurately implement multi-label cases for Equation 10 and 11 with only CGAT's description.

To verify the functionality of our implementation, we report the results of a full-supervised setting with the original loss (Equation 13) like CGAT paper, in Table 7. Our implementation of CGAT shows almost the same performance reported in the original paper. In addition, SuperGAT and CGAT showed almost similar performance in a full-supervised setting. Note that the original paper employs two hidden layers with hidden dimensions as 32 for Cora, 64 for CiteSeer, and three hidden layers with hidden dimensions 32 for PubMed, where models in our experiments are all two-layer with 64 features.

## A.7 WALL-CLOCK TIME EXPERIMENTAL SET-UP

To demonstrate our model's efficiency, we measure the mean wall-clock time of the entire training process of three runs using a single GPU (GeForce GTX 1080Ti). We compare our model with GAT (Veličković et al., 2018) and GAM (Stretcu et al., 2019). GAT is the basic model using a simpler attention mechanism than ours, and GAM is the state-of-the-art model using co-training with the auxiliary model.

For GAT and SuperGAT, we use our implementation (including hyperparameter settings) in Py-Torch (Paszke et al., 2019). For GAM, we adopt the code in TensorFlow (Abadi et al., 2015) from the authors [5] and choose GCN + GAM model, which showed the best performance. We retain the default settings in the code but use the hyperparameters reported in the paper, if possible. With this setting, GCN + GAM on PubMed is not finished after 24 hours; therefore, we manually early-stop the training at the best accuracy.

---

[5] `https://github.com/tensorflow/neural-structured-learning/tree/master/research/gam`

# B  RESULTS

## B.1  PROOF OF PROPOSITION

**Proposition 2.** *For $l + 1th$ GAT layer, if $\boldsymbol{W}$ and $\boldsymbol{a}$ are independent and identically drawn from zero-mean uniform distribution with variance $\sigma_w^2$ and $\sigma_a^2$ respectively, assuming that parameters are independent to input features $\boldsymbol{h}^l$ and elements of $\boldsymbol{h}^l$ are independent to each other,*

$$Var[e_{ij,GO}^{l+1}] = 2F^{l+1}\sigma_w^2\sigma_a^2\mathbb{E}(\|\boldsymbol{h}^l\|_2^2) \;\; and \;\; Var[e_{ij,DP}^{l+1}] \geq F^{l+1}\sigma_w^4\left(\tfrac{4}{5}\mathbb{E}\left(((\boldsymbol{h}_i^l)^\top\boldsymbol{h}_j^l)^2\right) + Var((\boldsymbol{h}_i^l)^\top\boldsymbol{h}_j^l)\right) \tag{15}$$

*Proof.* Let $\boldsymbol{h}' = \boldsymbol{W}\boldsymbol{h}^l$, then $\boldsymbol{h}'_{i,k} = \sum_{r=1}^{F^{l+1}}\boldsymbol{W}_{kr}\boldsymbol{h}_{i,r}^l$.

Note that,

$$e_{ij,GO}^{l+1} = \boldsymbol{a}^\top[\boldsymbol{h}_i'\|\boldsymbol{h}_j'] \quad and \quad e_{ij,DP}^{l+1} = \boldsymbol{h}_i'^\top\boldsymbol{h}_j' \tag{16}$$

First, we compute $\mathbb{E}(a^2)$ and $\mathbb{E}(h'^2)$.

$$\mathbb{E}(a^2) = \mathrm{Var}(a) + \mathbb{E}(a)^2 = \sigma_a^2 \tag{17}$$

$$\mathbb{E}(h_k'^2) = \mathbb{E}\left(\left(\sum_{r=1}^{F^{l+1}}\boldsymbol{W}_{kr}\boldsymbol{h}_{\cdot,r}^l\right)^2\right) \tag{18}$$

$$= \mathbb{E}\left(\sum_{r=1}^{F^{l+1}}\boldsymbol{W}_{kr}^2(\boldsymbol{h}_{\cdot,r}^l)^2\right) \tag{19}$$

$$= \mathbb{E}\left(\boldsymbol{W}_{k,\cdot}^2\right)\mathbb{E}\left(\sum_{r=1}^{F^{l+1}}(\boldsymbol{h}_{\cdot,r}^l)^2\right) \tag{20}$$

$$= \sigma_w^2\mathbb{E}\left(\|\boldsymbol{h}^l\|_2^2\right) \tag{21}$$

For the variance of $e_{ij,GO}^{l+1}$,

$$\mathrm{Var}(e_{ij,GO}^{l+1}) = \mathrm{Var}(\boldsymbol{a}^\top[\boldsymbol{h}_i'\|\boldsymbol{h}_j']) \tag{22}$$

$$= \mathrm{Var}\left(\sum_{r=1}^{F^{l+1}}(a_r\boldsymbol{h}_{i,r}' + a_{r+F^{l+1}}\boldsymbol{h}_{j,r}')\right) \tag{23}$$

$$= 2F^{l+1}\mathrm{Var}(ah') \tag{24}$$

$$= 2F^{l+1}\left(\mathbb{E}\left(a^2\right)\mathbb{E}\left(h'^2\right) - \mathbb{E}\left(a\right)^2\mathbb{E}\left(h'\right)^2\right) \tag{25}$$

$$= 2F^{l+1}\mathbb{E}\left(a^2\right)\mathbb{E}\left(h'^2\right) \tag{26}$$

$$= 2F^{l+1}\sigma_a^2\sigma_w^2\mathbb{E}\left(\|\boldsymbol{h}^l\|_2^2\right) \tag{27}$$

Now we compute $\mathbb{E}(h_i'h_j')$ and $\mathbb{E}(h_i'^2h_j'^2)$,

$$\mathbb{E}(\boldsymbol{h}_{i,k}'\boldsymbol{h}_{j,k}') = \mathbb{E}\left(\left(\sum_{r=1}^{F^{l+1}}\boldsymbol{W}_{kr}\boldsymbol{h}_{i,r}^l\right)\left(\sum_{r=1}^{F^{l+1}}\boldsymbol{W}_{kr}\boldsymbol{h}_{j,r}^l\right)\right) \tag{28}$$

$$= \mathbb{E}\left(\sum_{r=1}^{F^{l+1}}\boldsymbol{W}_{kr}^2\boldsymbol{h}_{i,r}^l\boldsymbol{h}_{j,r}^l\right) \tag{29}$$

$$= \mathbb{E}(\boldsymbol{W}_{k,\cdot}^2)\mathbb{E}\left(\sum_{r=1}^{F^{l+1}}\boldsymbol{h}_{i,r}^l\boldsymbol{h}_{j,r}^l\right) \tag{30}$$

$$= \sigma_w^2\mathbb{E}((\boldsymbol{h}_i^l)^\top\boldsymbol{h}_j^l) \tag{31}$$

$$\mathbb{E}(\boldsymbol{h}_{i,k}'^2 \boldsymbol{h}_{j,k}'^2) \tag{32}$$

$$= \mathbb{E}\left(\left(\sum_{r=1}^{F^{l+1}} \boldsymbol{W}_{kr}\boldsymbol{h}_{i,r}^l\right)^2 \left(\sum_{r=1}^{F^{l+1}} \boldsymbol{W}_{kr}\boldsymbol{h}_{j,r}^l\right)^2\right) \tag{33}$$

$$= \mathbb{E}(\boldsymbol{W}_{k,\cdot}^4)\mathbb{E}\left(\sum_{r=1}^{F^{l+1}} (\boldsymbol{h}_{i,r}^l \boldsymbol{h}_{j,r}^l)^2\right) + \mathbb{E}(\boldsymbol{W}_{k,\cdot}^2)^2 \mathbb{E}\left(\sum_{s\neq t}^{F^{l+1}} (\boldsymbol{h}_{i,s}^l \boldsymbol{h}_{j,t}^l)^2 + 2(\boldsymbol{h}_{i,s}^l \boldsymbol{h}_{j,s}^l)(\boldsymbol{h}_{i,t}^l \boldsymbol{h}_{j,t}^l)\right) \tag{34}$$

$$= \mathbb{E}(\boldsymbol{W}_{k,\cdot}^4)\mathbb{E}\left(\sum_{r=1}^{F^{l+1}} (\boldsymbol{h}_{i,r}^l \boldsymbol{h}_{j,r}^l)^2\right) + \mathbb{E}(\boldsymbol{W}_{k,\cdot}^2)^2 \mathbb{E}\left(\sum_r (\boldsymbol{h}_{i,r}^l)^2 \sum_r (\boldsymbol{h}_{j,r}^l)^2 - \sum_r (\boldsymbol{h}_{i,r}^l \boldsymbol{h}_{j,r}^l)^2\right) \tag{35}$$

$$+ 2\mathbb{E}(\boldsymbol{W}_{k,\cdot}^2)^2 \mathbb{E}\left(\left(\sum_r \boldsymbol{h}_{i,r}^l \boldsymbol{h}_{j,r}^l\right)^2 - \sum_r (\boldsymbol{h}_{i,r}^l \boldsymbol{h}_{j,r}^l)^2\right) \tag{36}$$

$$= \frac{9}{5}\sigma_w^4 \mathbb{E}\left(\sum_{r=1}^{F^{l+1}} (\boldsymbol{h}_{i,r}^l \boldsymbol{h}_{j,r}^l)^2\right) + \sigma_w^4 \mathbb{E}\left(\|\boldsymbol{h}_i^l\|_2^2 \|\boldsymbol{h}_j^l\|_2^2 - 3\sum_{r=1}^{F^{l+1}} (\boldsymbol{h}_{i,r}^l \boldsymbol{h}_{j,r}^l)^2\right) + 2\sigma_w^4 \mathbb{E}\left(((\boldsymbol{h}_i^l)^\top \boldsymbol{h}_j^l)^2\right) \tag{37}$$

$$= \sigma_w^4 \mathbb{E}\left(\|\boldsymbol{h}_i^l\|_2^2 \|\boldsymbol{h}_j^l\|_2^2 - \frac{6}{5}\sum_{r=1}^{F^{l+1}} (\boldsymbol{h}_{i,r}^l \boldsymbol{h}_{j,r}^l)^2 + 2((\boldsymbol{h}_i^l)^\top \boldsymbol{h}_j^l)^2\right) \tag{38}$$

$$= \sigma_w^4 \mathbb{E}\left(\frac{4}{5}((\boldsymbol{h}_i^l)^\top \boldsymbol{h}_j^l)^2 + \frac{1}{10}\sum_{s\neq t} \left((\boldsymbol{h}_{i,s}^l \boldsymbol{h}_{j,t}^l + \boldsymbol{h}_{i,t}^l \boldsymbol{h}_{j,s}^l)^2 + 8(\boldsymbol{h}_{i,s}^l \boldsymbol{h}_{j,t}^l)^2\right)\right) + \sigma_w^4 \mathbb{E}\left(((\boldsymbol{h}_i^l)^\top \boldsymbol{h}_j^l)^2\right) \tag{39}$$

Note that for the zero-mean uniform distribution $\mathcal{U}(-u, u)$ with variance $\sigma_w^2$,

$$\mathbb{E}(\boldsymbol{W}_{\cdot,\cdot}^2) = \mathrm{Var}(\boldsymbol{W}_{\cdot,\cdot}) + \mathbb{E}(\boldsymbol{W}_{\cdot,\cdot})^2 = \sigma_w^2 \tag{40}$$

$$\sigma_w^2 = \frac{1}{12}(u - (-u))^2 = \frac{1}{3}u^2 \tag{41}$$

$$\mathbb{E}(\boldsymbol{W}_{\cdot,\cdot}^4) = \frac{1}{5}\sum_{r=0}^4 (-u)^r u^{4-r} = \frac{1}{5}u^4 = \frac{9}{5}\sigma_w^4 \tag{42}$$

For the variance of $e_{ij,\mathrm{DP}}^{l+1}$,

$$\mathrm{Var}\left(e_{ij,\mathrm{DP}}^{l+1}\right) \tag{43}$$

$$= \mathrm{Var}\left(\boldsymbol{h}_i'^\top \boldsymbol{h}_j'\right) \tag{44}$$

$$= \mathrm{Var}\left(\sum_{r=1}^{F^{l+1}} \boldsymbol{h}_{i,r}' \boldsymbol{h}_{j,r}'\right) \tag{45}$$

$$= F^{l+1}\mathrm{Var}\left(h_i' h_j'\right) \tag{46}$$

$$= F^{l+1}\left(\mathbb{E}((h_i' h_j')^2) - \mathbb{E}(h_i' h_j')^2\right) \tag{47}$$

$$= F^{l+1}\sigma_w^4 \mathbb{E}\left(\frac{4}{5}((\boldsymbol{h}_i^l)^\top \boldsymbol{h}_j^l)^2 + \frac{1}{10}\sum_{s\neq t} \left((\boldsymbol{h}_{i,s}^l \boldsymbol{h}_{j,t}^l + \boldsymbol{h}_{i,t}^l \boldsymbol{h}_{j,s}^l)^2 + 8(\boldsymbol{h}_{i,s}^l \boldsymbol{h}_{j,t}^l)^2\right)\right) \tag{48}$$

$$+ F^{l+1}\sigma_w^4 \left(\mathbb{E}\left(((\boldsymbol{h}_i^l)^\top \boldsymbol{h}_j^l)^2\right) - \mathbb{E}((\boldsymbol{h}_i^l)^\top \boldsymbol{h}_j^l)^2\right) \tag{49}$$

$$\geq F^{l+1}\sigma_w^4 \left(\frac{4}{5}\mathbb{E}\left(((\boldsymbol{h}_i^l)^\top \boldsymbol{h}_j^l)^2\right) + \mathrm{Var}((\boldsymbol{h}_i^l)^\top \boldsymbol{h}_j^l)\right) \tag{50}$$

$$\square$$

Table 8: Summary of classification accuracies (of 5 runs) for synthetic datasets.

| Avg. degree | Homophily | | | | | | | | |
|---|---|---|---|---|---|---|---|---|---|
| GCN | 0.1 | 0.2 | 0.3 | 0.4 | 0.5 | 0.6 | 0.7 | 0.8 | 0.9 |
| 1 | 32.7 ± 1.1 | 35.1 ± 0.4 | 34.7 ± 0.4 | 40.8 ± 0.4 | 43.4 ± 0.3 | 46.2 ± 0.5 | 49.7 ± 0.5 | 54.8 ± 0.9 | 56.2 ± 0.3 |
| 1.5 | 28.7 ± 0.5 | 30.0 ± 0.6 | 34.6 ± 0.4 | 40.8 ± 0.4 | 41.7 ± 0.4 | 46.9 ± 0.5 | 52.1 ± 0.6 | 54.5 ± 0.5 | 63.6 ± 0.5 |
| 2.5 | 24.6 ± 0.2 | 27.9 ± 0.9 | 31.0 ± 0.3 | 39.7 ± 0.6 | 44.0 ± 0.6 | 51.4 ± 0.8 | 56.6 ± 0.5 | 65.2 ± 0.6 | 74.5 ± 0.4 |
| 3.5 | 25.8 ± 0.8 | 30.5 ± 0.8 | 31.9 ± 0.7 | 38.4 ± 0.4 | 45.0 ± 1.0 | 49.8 ± 0.5 | 60.8 ± 0.7 | 70.3 ± 0.3 | 79.5 ± 0.3 |
| 5 | 25.0 ± 0.3 | 26.3 ± 0.5 | 33.0 ± 1.1 | 39.8 ± 0.7 | 50.2 ± 0.5 | 58.6 ± 1.1 | 68.8 ± 0.3 | 78.0 ± 1.9 | 88.5 ± 0.2 |
| 7.5 | 26.1 ± 0.5 | 29.8 ± 0.6 | 34.0 ± 0.6 | 43.8 ± 0.9 | 52.0 ± 0.4 | 70.7 ± 3.2 | 74.6 ± 2.4 | 88.6 ± 0.8 | 95.0 ± 0.2 |
| 10 | 25.4 ± 0.6 | 29.2 ± 0.5 | 37.1 ± 0.5 | 47.6 ± 1.0 | 58.1 ± 0.9 | 73.1 ± 2.8 | 85.3 ± 2.6 | 92.7 ± 1.1 | 97.9 ± 0.5 |
| 12.5 | 24.0 ± 0.4 | 27.6 ± 0.5 | 39.6 ± 0.8 | 48.9 ± 0.4 | 65.8 ± 3.1 | 77.8 ± 2.4 | 88.5 ± 1.5 | 95.0 ± 0.9 | 99.2 ± 0.3 |
| 15 | 22.1 ± 0.5 | 28.2 ± 0.6 | 44.0 ± 0.7 | 53.0 ± 0.5 | 69.9 ± 4.7 | 79.1 ± 2.0 | 92.2 ± 1.8 | 98.0 ± 0.4 | 99.6 ± 0.3 |
| 20 | 24.3 ± 0.6 | 30.9 ± 0.7 | 41.9 ± 0.9 | 58.1 ± 1.2 | 75.0 ± 2.9 | 86.7 ± 1.1 | 96.1 ± 0.3 | 98.9 ± 0.1 | 99.8 ± 0.1 |
| 25 | 26.6 ± 1.0 | 31.1 ± 0.3 | 44.9 ± 0.2 | 62.4 ± 0.7 | 80.1 ± 2.6 | 91.0 ± 1.5 | 97.5 ± 0.5 | 99.5 ± 0.2 | 100.0 ± 0.0 |
| 32.5 | 26.3 ± 0.7 | 33.8 ± 0.7 | 51.8 ± 0.8 | 67.5 ± 2.9 | 83.5 ± 1.9 | 95.7 ± 0.3 | 98.4 ± 0.2 | 100.0 ± 0.0 | 100.0 ± 0.0 |
| 40 | 23.7 ± 0.5 | 34.2 ± 0.5 | 53.4 ± 0.4 | 72.8 ± 1.5 | 87.6 ± 0.9 | 96.1 ± 0.6 | 99.7 ± 0.1 | 99.9 ± 0.0 | 100.0 ± 0.0 |
| 50 | 25.0 ± 1.1 | 36.4 ± 1.0 | 55.1 ± 0.5 | 82.7 ± 3.0 | 91.2 ± 0.7 | 98.6 ± 0.4 | 99.8 ± 0.0 | 99.9 ± 0.0 | 100.0 ± 0.0 |
| 75 | 25.9 ± 0.6 | 40.6 ± 0.6 | 68.0 ± 1.0 | 87.9 ± 2.6 | 97.1 ± 0.6 | 99.6 ± 0.1 | 100.0 ± 0.0 | 100.0 ± 0.0 | 100.0 ± 0.0 |
| 100 | 25.1 ± 0.5 | 42.0 ± 0.9 | 72.4 ± 1.4 | 92.6 ± 0.4 | 98.6 ± 0.2 | 100.0 ± 0.0 | 100.0 ± 0.0 | 100.0 ± 0.0 | 100.0 ± 0.0 |
| GAT$_{GO}$ | 0.1 | 0.2 | 0.3 | 0.4 | 0.5 | 0.6 | 0.7 | 0.8 | 0.9 |
| 1 | 33.7 ± 0.9 | 36.7 ± 0.8 | 35.2 ± 1.2 | 40.1 ± 0.7 | 43.0 ± 0.8 | 46.3 ± 0.6 | 49.4 ± 0.5 | 53.8 ± 1.4 | 56.1 ± 0.5 |
| 1.5 | 28.8 ± 1.0 | 33.0 ± 0.5 | 34.8 ± 0.9 | 40.7 ± 0.4 | 42.2 ± 0.9 | 46.7 ± 1.0 | 52.2 ± 0.8 | 53.6 ± 1.4 | 62.8 ± 0.5 |
| 2.5 | 28.8 ± 1.2 | 30.6 ± 0.6 | 32.3 ± 0.5 | 40.1 ± 0.8 | 44.0 ± 1.3 | 52.2 ± 0.6 | 55.5 ± 1.4 | 64.6 ± 1.3 | 73.1 ± 0.6 |
| 3.5 | 28.0 ± 0.4 | 33.4 ± 1.6 | 33.5 ± 0.6 | 40.1 ± 1.0 | 46.2 ± 0.6 | 50.8 ± 0.6 | 62.5 ± 0.6 | 71.7 ± 0.6 | 78.0 ± 0.5 |
| 5 | 28.0 ± 1.3 | 30.0 ± 0.8 | 36.9 ± 0.6 | 42.1 ± 0.4 | 53.6 ± 0.7 | 61.0 ± 1.1 | 70.4 ± 0.5 | 78.7 ± 1.2 | 87.5 ± 0.6 |
| 7.5 | 30.0 ± 0.9 | 31.6 ± 2.0 | 40.4 ± 0.5 | 45.0 ± 1.0 | 59.5 ± 0.6 | 72.3 ± 1.0 | 79.4 ± 0.6 | 90.4 ± 0.5 | 95.0 ± 0.5 |
| 10 | 31.0 ± 1.4 | 34.0 ± 1.2 | 43.1 ± 0.6 | 54.6 ± 0.5 | 66.1 ± 0.6 | 77.4 ± 1.5 | 90.6 ± 1.4 | 94.7 ± 0.7 | 98.4 ± 0.5 |
| 12.5 | 29.8 ± 1.4 | 35.5 ± 1.7 | 47.3 ± 0.9 | 58.6 ± 1.7 | 72.1 ± 1.4 | 86.3 ± 0.5 | 90.7 ± 0.7 | 96.2 ± 0.3 | 98.9 ± 0.2 |
| 15 | 31.9 ± 1.6 | 34.6 ± 1.6 | 49.5 ± 0.9 | 62.0 ± 0.7 | 77.2 ± 1.5 | 85.9 ± 0.3 | 94.4 ± 0.9 | 98.5 ± 0.5 | 99.4 ± 0.1 |
| 20 | 34.4 ± 1.8 | 38.3 ± 1.6 | 54.1 ± 1.9 | 70.0 ± 1.1 | 83.9 ± 1.0 | 93.7 ± 0.1 | 97.5 ± 0.3 | 99.0 ± 0.3 | 99.7 ± 0.0 |
| 25 | 35.8 ± 2.0 | 42.0 ± 2.4 | 57.7 ± 0.6 | 77.7 ± 0.9 | 87.9 ± 1.2 | 95.0 ± 0.7 | 98.7 ± 0.6 | 99.6 ± 0.3 | 99.9 ± 0.1 |
| 32.5 | 37.4 ± 1.2 | 44.7 ± 1.1 | 66.5 ± 1.9 | 79.9 ± 1.2 | 91.4 ± 1.4 | 98.0 ± 0.5 | 99.0 ± 0.3 | 99.8 ± 0.1 | 100.0 ± 0.0 |
| 40 | 37.5 ± 2.0 | 45.1 ± 0.9 | 66.5 ± 1.1 | 85.7 ± 1.5 | 93.5 ± 1.0 | 97.6 ± 0.6 | 99.5 ± 0.1 | 99.9 ± 0.1 | 99.9 ± 0.1 |
| 50 | 38.7 ± 1.8 | 49.5 ± 2.3 | 68.9 ± 2.5 | 89.5 ± 0.8 | 96.0 ± 0.8 | 99.0 ± 0.4 | 99.7 ± 0.2 | 99.8 ± 0.1 | 100.0 ± 0.0 |
| 75 | 39.6 ± 3.0 | 53.5 ± 1.7 | 77.6 ± 2.6 | 92.8 ± 2.3 | 98.0 ± 1.2 | 99.5 ± 0.3 | 99.8 ± 0.2 | 100.0 ± 0.0 | 100.0 ± 0.0 |
| 100 | 41.3 ± 1.2 | 56.6 ± 1.4 | 81.1 ± 3.5 | 95.9 ± 1.2 | 99.0 ± 0.4 | 99.8 ± 0.1 | 99.9 ± 0.1 | 100.0 ± 0.1 | 100.0 ± 0.0 |
| SuperGAT$_{SD}$ | 0.1 | 0.2 | 0.3 | 0.4 | 0.5 | 0.6 | 0.7 | 0.8 | 0.9 |
| 1 | 38.5 ± 1.0 | 40.5 ± 1.5 | 39.5 ± 1.6 | 42.3 ± 0.9 | 44.0 ± 0.5 | 47.1 ± 0.5 | 50.1 ± 1.0 | 53.0 ± 0.3 | 55.1 ± 1.0 |
| 1.5 | 36.7 ± 1.2 | 37.8 ± 1.8 | 42.0 ± 0.4 | 42.9 ± 0.8 | 45.0 ± 0.7 | 47.8 ± 0.5 | 52.9 ± 0.7 | 54.7 ± 1.2 | 63.0 ± 0.4 |
| 2.5 | 35.7 ± 1.8 | 37.3 ± 1.6 | 39.2 ± 1.4 | 47.8 ± 1.3 | 49.0 ± 0.3 | 56.1 ± 0.5 | 59.4 ± 0.4 | 65.9 ± 0.8 | 70.6 ± 1.2 |
| 3.5 | 35.7 ± 1.0 | 39.2 ± 1.9 | 41.4 ± 0.5 | 44.0 ± 0.7 | 51.0 ± 0.3 | 55.0 ± 1.2 | 65.1 ± 1.1 | 73.5 ± 1.0 | 78.1 ± 0.6 |
| 5 | 37.0 ± 1.5 | 39.0 ± 1.5 | 45.0 ± 1.7 | 48.1 ± 1.0 | 57.5 ± 0.6 | 66.1 ± 0.9 | 74.4 ± 0.2 | 81.0 ± 1.0 | 88.3 ± 0.8 |
| 7.5 | 36.8 ± 1.5 | 38.7 ± 0.7 | 47.3 ± 0.5 | 51.1 ± 0.8 | 61.2 ± 0.6 | 75.5 ± 0.5 | 82.3 ± 0.9 | 91.0 ± 0.2 | 95.2 ± 0.3 |
| 10 | 39.4 ± 1.1 | 42.3 ± 0.7 | 50.5 ± 1.2 | 58.6 ± 0.3 | 70.0 ± 0.5 | 80.1 ± 0.5 | 90.8 ± 0.3 | 95.5 ± 0.6 | 98.7 ± 0.3 |
| 12.5 | 38.5 ± 1.0 | 42.0 ± 0.7 | 50.5 ± 0.6 | 62.4 ± 0.5 | 73.6 ± 0.4 | 86.8 ± 0.4 | 92.9 ± 0.4 | 98.0 ± 0.6 | 99.3 ± 0.3 |
| 15 | 40.4 ± 1.1 | 42.8 ± 0.7 | 53.6 ± 0.3 | 67.4 ± 0.4 | 79.7 ± 0.3 | 89.0 ± 1.0 | 96.3 ± 0.3 | 99.1 ± 0.2 | 99.6 ± 0.3 |
| 20 | 37.8 ± 0.8 | 42.1 ± 0.9 | 56.9 ± 0.9 | 70.0 ± 0.5 | 86.1 ± 0.8 | 95.6 ± 0.3 | 99.1 ± 0.1 | 99.6 ± 0.2 | 99.7 ± 0.1 |
| 25 | 40.0 ± 1.0 | 48.8 ± 1.2 | 59.5 ± 0.5 | 78.7 ± 0.2 | 90.7 ± 0.2 | 97.5 ± 0.1 | 99.4 ± 0.1 | 100.0 ± 0.0 | 99.9 ± 0.1 |
| 32.5 | 39.7 ± 1.1 | 48.5 ± 0.7 | 69.4 ± 0.4 | 82.7 ± 0.5 | 94.3 ± 0.2 | 99.3 ± 0.1 | 99.7 ± 0.1 | 99.9 ± 0.1 | 99.9 ± 0.0 |
| 40 | 44.2 ± 1.1 | 48.7 ± 1.3 | 69.2 ± 0.7 | 88.3 ± 0.4 | 97.1 ± 0.1 | 99.7 ± 0.1 | 99.9 ± 0.0 | 99.8 ± 0.2 | 100.0 ± 0.0 |
| 50 | 44.3 ± 0.7 | 53.2 ± 0.6 | 73.4 ± 1.2 | 91.2 ± 0.4 | 97.7 ± 0.2 | 100.0 ± 0.0 | 100.0 ± 0.0 | 100.0 ± 0.0 | 100.0 ± 0.0 |
| 75 | 44.8 ± 0.7 | 56.1 ± 0.9 | 82.7 ± 0.6 | 95.7 ± 0.2 | 98.8 ± 0.2 | 99.9 ± 0.1 | 100.0 ± 0.0 | 100.0 ± 0.0 | 100.0 ± 0.0 |
| 100 | 43.1 ± 1.3 | 60.7 ± 0.7 | 87.4 ± 0.3 | 98.3 ± 0.1 | 99.9 ± 0.0 | 100.0 ± 0.0 | 100.0 ± 0.0 | 100.0 ± 0.0 | 100.0 ± 0.0 |
| SuperGAT$_{MX}$ | 0.1 | 0.2 | 0.3 | 0.4 | 0.5 | 0.6 | 0.7 | 0.8 | 0.9 |
| 1 | 33.4 ± 0.7 | 36.3 ± 0.8 | 35.4 ± 0.7 | 40.5 ± 0.3 | 43.4 ± 0.6 | 47.0 ± 0.8 | 50.0 ± 0.5 | 54.2 ± 1.1 | 56.4 ± 0.4 |
| 1.5 | 28.8 ± 0.7 | 31.5 ± 1.0 | 36.0 ± 0.8 | 41.1 ± 1.1 | 41.3 ± 0.6 | 47.5 ± 0.8 | 52.3 ± 0.7 | 54.8 ± 0.5 | 63.4 ± 0.3 |
| 2.5 | 26.6 ± 0.9 | 30.5 ± 1.3 | 33.6 ± 0.8 | 42.7 ± 1.5 | 44.7 ± 0.5 | 52.2 ± 0.5 | 56.5 ± 0.8 | 66.7 ± 0.8 | 74.5 ± 0.7 |
| 3.5 | 27.7 ± 0.9 | 33.9 ± 1.7 | 37.9 ± 0.7 | 42.4 ± 0.6 | 49.7 ± 0.8 | 54.2 ± 1.1 | 65.3 ± 1.1 | 73.2 ± 0.7 | 79.4 ± 1.5 |
| 5 | 28.4 ± 2.1 | 33.4 ± 0.8 | 40.9 ± 0.8 | 46.2 ± 0.4 | 56.4 ± 0.2 | 64.1 ± 1.8 | 74.6 ± 1.4 | 83.1 ± 1.0 | 90.6 ± 0.9 |
| 7.5 | 31.1 ± 0.7 | 35.0 ± 0.9 | 46.5 ± 1.1 | 51.9 ± 0.7 | 64.3 ± 1.5 | 75.2 ± 1.6 | 82.1 ± 1.3 | 92.6 ± 1.8 | 95.8 ± 0.2 |
| 10 | 32.6 ± 1.7 | 41.5 ± 1.4 | 48.9 ± 1.2 | 61.7 ± 1.1 | 70.4 ± 1.3 | 81.3 ± 1.1 | 91.3 ± 0.7 | 95.1 ± 0.8 | 99.2 ± 0.3 |
| 12.5 | 34.2 ± 3.1 | 41.1 ± 0.8 | 53.3 ± 1.0 | 65.9 ± 1.1 | 77.2 ± 1.4 | 88.1 ± 0.7 | 92.6 ± 2.6 | 97.5 ± 0.9 | 99.3 ± 0.2 |
| 15 | 36.4 ± 1.3 | 39.9 ± 2.1 | 53.8 ± 1.5 | 68.5 ± 0.8 | 81.0 ± 0.4 | 90.2 ± 1.0 | 96.1 ± 0.9 | 99.2 ± 0.2 | 99.6 ± 0.1 |
| 20 | 35.2 ± 3.1 | 41.0 ± 2.1 | 62.3 ± 1.2 | 73.9 ± 0.9 | 86.3 ± 0.6 | 95.2 ± 1.3 | 98.6 ± 0.7 | 99.6 ± 0.1 | 99.9 ± 0.2 |
| 25 | 36.8 ± 1.9 | 46.6 ± 1.7 | 62.7 ± 1.1 | 81.9 ± 1.7 | 90.7 ± 0.8 | 96.8 ± 1.5 | 99.3 ± 0.3 | 99.8 ± 0.2 | 100.0 ± 0.0 |
| 32.5 | 36.7 ± 0.8 | 46.8 ± 0.5 | 68.9 ± 0.6 | 83.8 ± 0.7 | 93.7 ± 1.0 | 98.4 ± 0.7 | 99.8 ± 0.1 | 100.0 ± 0.0 | 100.0 ± 0.0 |
| 40 | 39.8 ± 2.4 | 48.8 ± 1.7 | 72.1 ± 0.8 | 87.5 ± 1.7 | 96.6 ± 0.8 | 98.9 ± 0.3 | 99.9 ± 0.1 | 100.0 ± 0.0 | 100.0 ± 0.0 |
| 50 | 42.1 ± 1.3 | 53.6 ± 2.3 | 75.1 ± 1.9 | 92.7 ± 1.0 | 97.1 ± 0.9 | 99.6 ± 0.2 | 100.0 ± 0.0 | 100.0 ± 0.0 | 100.0 ± 0.0 |
| 75 | 41.9 ± 1.1 | 55.9 ± 2.4 | 80.8 ± 1.2 | 95.0 ± 1.1 | 99.5 ± 0.2 | 99.9 ± 0.1 | 100.0 ± 0.0 | 100.0 ± 0.0 | 100.0 ± 0.0 |
| 100 | 39.6 ± 2.0 | 59.2 ± 1.8 | 84.2 ± 3.1 | 96.9 ± 0.7 | 99.7 ± 0.1 | 100.0 ± 0.0 | 100.0 ± 0.0 | 100.0 ± 0.0 | 100.0 ± 0.0 |

## B.2 Full Result of Synthetic Graph Experiments

In Table 8, we report all results of synthetic graph experiments. We experiment on a total of 144 synthetic graphs controlling 9 homophily (0.1, 0.2, ..., 0.9) and 16 average degree (1, 1.5, 2.5, 3.5, 5, 7.5, 10, 12.5, 15, 20, 25, 32.5, 40, 50, 75, 100).

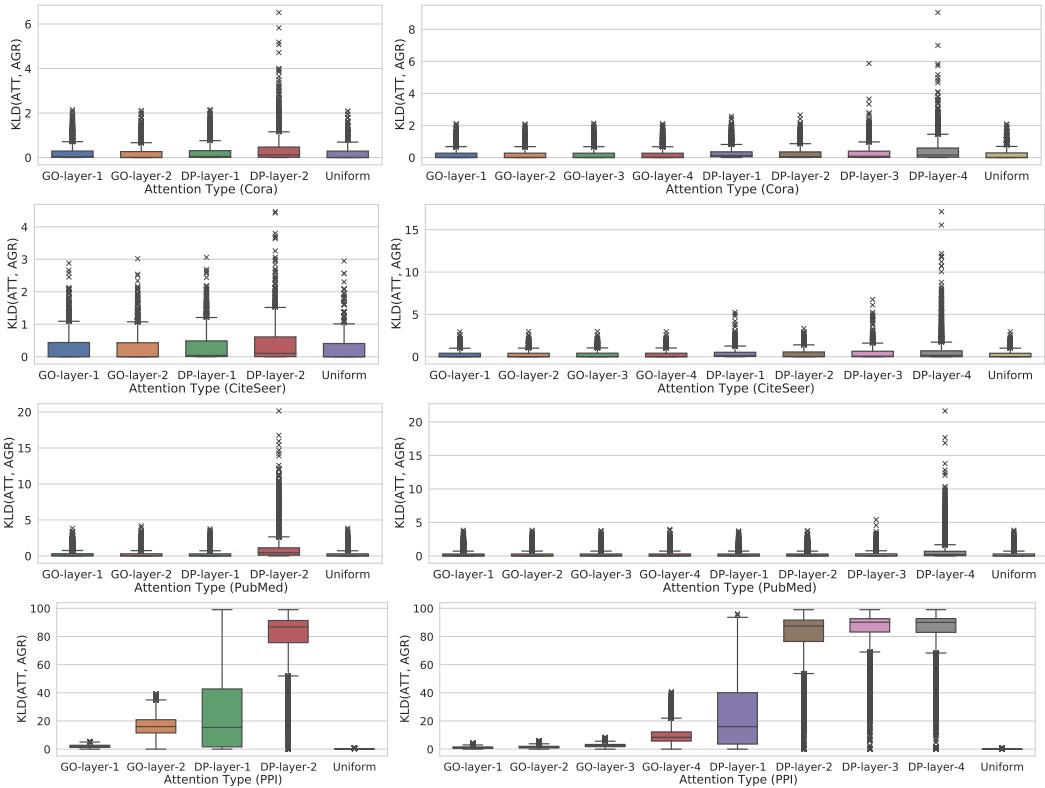

Figure 8: Distribution of KL divergence between normalized attention and label-agreement on all nodes and layers for Cora, CiteSeer, PubMed, and PPI (Left: two-layer GAT, Right: four-layer GAT).

Table 9: Mean wall-clock time (seconds) of three runs of the training process on real-world datasets.

| Model | Cora | CiteSeer | PubMed |
|---|---|---|---|
| GAT | $11.3 \pm 2.7$ | $20.4 \pm 6.7$ | $21.1 \pm 2.0$ |
| GCN + GAM | $709.3 \pm 235.9$ | $1099.3 \pm 812.5$ | $6923.3 \pm 7042.0$ |
| SuperGAT$_{\text{MX}}$ | $30.8 \pm 0.5$ | $19.3 \pm 1.1$ | $151.4 \pm 11.4$ |
| SuperGAT$_{\text{MX}}$+ MPNS | $20.8 \pm 0.2$ | $15.2 \pm 0.1$ | $84.6 \pm 0.9$ |

### B.3 LABEL-AGREEMENT STUDY FOR OTHER DATASETS AND DEEPER MODELS

In Figure 8, we draw box plots of KL divergence between attention distribution and label agreement distribution for all nodes and layers of two-layer GATs and four-layer GATs. As shown in the paper, we can see that DP attention does not capture label-agreement rather than GO attention. Also, the degree of this phenomenon becomes stronger as the layer goes down.

### B.4 WALL-CLOCK TIME RESULT

In Table 9, we report the mean wall-clock time (over three runs) of the training of GAT, GAM, and SuperGAT$_{\text{MX}}$. In SuperGAT, we find that negative sampling of edges is the bottleneck of training. So, we additionally implement SuperGAT$_{\text{MX}}$+ MPNS, which employs multi-processing when sampling negative edges. There are three observations in this experiment. GCN + GAM is highly time-intensive in the training stage ($\times 53.9 - \times 328.1$ versus GAT) for all datasets. Compared to GAT, our model needs $\times 2.7$ more training time for Cora and $\times 7.2$ for PubMed, and we reduce the time by applying multi-processing to negative sampling ($\times 1.8$ for Cora and $\times 4.0$ for PubMed). For CiteSeer, we can see that SuperGAT$_{\text{MX}}$ ends faster than GAT because of faster convergence and fewer epochs.

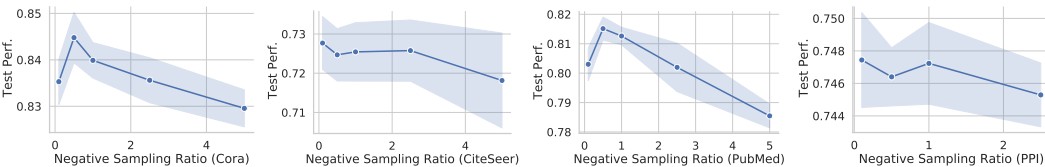

Figure 9: Test performance on node classification against the mixing coefficient $\lambda_E$ for SuperGAT$_{\text{MX}}$ (Cora, CiteSeer, PubMed) and SuperGAT$_{\text{SD}}$ (PPI).

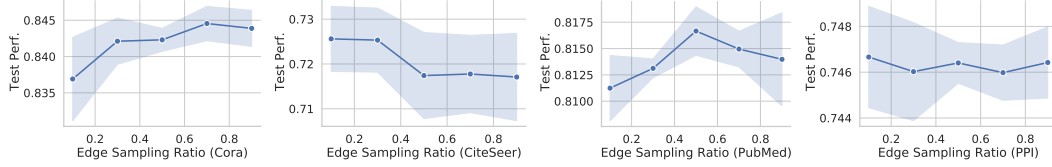

Figure 10: Test performance on node classification against the negative sampling ratio $p_n$ for SuperGAT$_{\text{MX}}$ (Cora, CiteSeer, PubMed) and SuperGAT$_{\text{SD}}$ (PPI).

Figure 11: Test performance on node classification against the edge sampling ratio $p_e$ for SuperGAT$_{\text{MX}}$ (Cora, CiteSeer, PubMed) and SuperGAT$_{\text{SD}}$ (PPI).

## B.5 SENSITIVITY ANALYSIS OF HYPER-PARAMETERS

We analyze sensitivity of mixing coefficient of losses $\lambda_E$, negative sampling ratio $p_n$, and edge sampling ratio $p_e$. We plot mean node classification performance (over 5 runs) against each hyper-parameter in Figure 9, 10, and 11 respectively. We use the best model for each dataset: SuperGAT$_{\text{MX}}$ for citation networks and SuperGAT$_{\text{SD}}$ for PPI.

For $\lambda_E$, there is a specific range that maximizes test performance in all datasets. Performance on PPI is the largest when $\lambda_E$ is $10^{-3}$, but the difference is relatively small comparing to others. We observe that there is an optimal level of the edge supervision for each dataset, and using too large $\lambda_E$ degrades node classification performance.

For $p_n$, using too many negative samples has been shown to decrease performance. The optimal number of negative samples is different for each dataset, and all are less than the number of positive samples ($p_n < 1.0$). Note that as $p_n$ increases, the required GPU memory also increases. When $p_n = 5.0$, the model and data for PPI could not be accommodated by one single GPU (GeForce GTX 1080Ti).

When $p_e$ changes, the performance also changes, but the pattern is different by datasets. For Cora and PubMed, the performance against $p_e$ shows the convex curve. Performance for CiteSeer generally decreases as $p_e$ increases, but there are intervals the performance change of which is nearly zero. In the case of PPI, there are no noticeable changes against $p_e$.

