# OpenReview forum: "How to Find Your Friendly Neighborhood: Graph Attention Design with Self-Supervision"
_ICLR.cc/2021/Conference — ICLR 2021 Poster_

### Official Review · AnonReviewer3 · 2020-10-24
**interesting method for graph attention network; thorough experiments; need to add a picture describing the general framework**

**Rating:** 7
**Confidence:** 3

**Review:**

********Summary
In this paper, they introduced self-supervised graph attention network (SuperGAT), which is claimed to perform well in noisy graphs. They used information in the edges as an indicator of importance of relations in the graph, then they learn the relational importance using self-supervised attention. After learning the attentions values using their self-supervised method, they can predict the likelihood of an edge between nodes. They worked on two popular attention mechanisms: GO and DP, and showed in their experiments DP has better performance than GO for link prediction task. And Go has better performance in label-agreements between nodes. The other question they answered in their experiments was what the best attention model is to choose. They introduce a recipe based on two graph characteristics: homophily and average degree.

********Positives
- One thing I liked about this paper was thorough and neat experiments. I enjoyed the way they designed their experiments by mentioning several important questions followed by their answered backed up with their experiments. They used two attention mechanism as the base, then applied their method on these two methods for link prediction and label-agreements tasks and compare their results.

- I also liked that they examined their recommendation for the choice of attention model on real world datasets, and their answer for real-word data was almost similar to synthetics data.


- The paper was well-organized and well-written. They clearly explained their method.

********Notes
- I would recommend adding a picture showing their architecture and compare it with other two attention models

- I sort-of understand the reading as to why "GO learns label-agreement better than DP." Based on the argument on page 6. A strong argument would be helpful to explain why "DP predicts edge presence better than GO."

- (minor note:)No need the parenthesis in this sentence line 8: "Interestingly, for datasets (CS, Physics, Cora-ML, and Flickr) in which"

********Reason to accept
I am in general positive about this paper. The innovation is not significant; however, their experiments were interesting, and they prove how well their method works well empirically. I think this research will be useful for people in this area.

******* After Rebuttal
I have read the author's response

---

> ### Author Response · Authors · 2020-11-16
> **Response to R3**
>
> Thank you for your review and positive feedback.
>
> > I would recommend adding a picture showing their architecture and compare it with other two attention models
>
> We added a figure of the model in the revised paper. Figure 1 contains the attention mechanism of all variants of SuperGAT and the original GAT.
>
> > No need the parenthesis in this sentence line 8: "Interestingly, for datasets (CS, Physics, Cora-ML, and Flickr) in which"
>
> Thank you R3. We fixed this issue.

---

### Official Review · AnonReviewer1 · 2020-10-27
**The authors present a new self-supervised attention mechanism for graph neural networks and conduct extensive experiments. The analysis and the design however lack some depth and it is uncertain how much this work enhances our understanding (or ability) to use attention in graphs.**

**Rating:** 5
**Confidence:** 4

**Review:**

This paper proposes a new attention mechanism SuperGAT (with various flavours) for graph neural networks that is self-supervised. They exploit the presence/absence of an edge between a pair of nodes to guide the attention. The authors then make the observation that the homophily and average degree of a graph influence the design of the attention mechanism.  Extensive experiments are shown, where the various versions of SuperGAT are tested on 17 real-world dataset and many synthetic ones, and these results are compared against other state-of-the-art models (including the original GAT work).

The paper is well-written and it is clear that the authors have done extensive experimentation to test their hypothesis.
However, the paper has some weaknesses that I try to summarize below.
- To obtain SuperGAT, the authors have made some tweaks made to the original GAT formulation. These tweaks are minor and are not surprising or inspired from a deep/novel insight.

- The choice of studying two graph properties homophily and average degree seem arbitrary. What is the reasoning behind these properties? Were there other properties (e.g. diameter, degree sequence, ...etc) that the authors have studied that did not yield good results? This is particularly of interest due to the fact that experiments do not entirely support/explain the importance of these two properties, such as in the case of Flickr and Crocodile datasets.

- The above reasons would still not be a major disadvantage had the experimental results shown strong superiority of SuperGAT over previous models. In all the experiments that the authors perform, SuperGAT’s performance is only slightly better than older models (e.g. in Table 2, difference between an accuracy of 72.5 vs 72.6 can hardly be considered superior). And even then, SuperGAT does not have the best performance across all datasets.

Proposition 1 and its proof are interesting contributions of this work, but they may not be enough. Perhaps instead of concentrating on performance superiority, the authors can look at the explainability aspect of their proposed architecture and look deeper into other graph properties that may guide the design/use of self-supervision.

Another minor comment: on Page 3, the authors say “if the number of nodes is large, it is not efficient to use all possible negative cases”. The number of negative cases is a function of the number of edges and not the number of nodes. If a graph contains all possible edges, then the number of all possible negative cases is zero, no matter how many nodes there are.

---

> ### Author Response · Authors · 2020-11-16
> **Response to R1**
>
> Thank you for your review and constructive feedback. We addressed two first bullets in the [general response](https://openreview.net/forum?id=Wi5KUNlqWty&noteId=iMkhL_qmEBe). We provide answers to your questions below.
>
> > In all the experiments that the authors perform, SuperGAT’s performance is only slightly better than older models (e.g. in Table 2, difference between an accuracy of 72.5 vs 72.6 can hardly be considered superior). And even then, SuperGAT does not have the best performance across all datasets.
>
> While SuperGAT does not outperform all baselines for all datasets, self-supervision contributes to a statistically significant improvement in 12 out of 17 real-world datasets. Moreover, inconsistency in performance improvement is shown in recent GNN literature including GAT, and in this paper, we reveal how that inconsistency can be explained by focusing on the characteristics of the input graph (RQ 3-4).
>
> > The number of negative cases is a function of the number of edges and not the number of nodes. If a graph contains all possible edges, then the number of all possible negative cases is zero, no matter how many nodes there are.
>
> The number of negative edges is a function of both the number of edges and nodes: $|V|^2 - |E|$ for $G = (V, E)$. SuperGAT is capable of modeling graphs that are sparse with a sufficiently large number of negative samples (i.e., $|V \times V| \gg |E|$), but this is generally not a problem because most real-world graphs are sparse [2]. For dense (or fully connected) graphs, it would be challenging to draw negative samples. These are a special class of graphs and rarely seen in the datasets in the graph neural network research community. We discuss this limitation in the paper.
>
> [2] Chung, Fan. "Graph theory in the information age." Notices of the AMS 57.6 (2010): 726-732.

---

### Official Review · AnonReviewer2 · 2020-10-28
**Recommendation for Acceptance**

**Rating:** 8
**Confidence:** 5

**Review:**

Summary:
===========
The paper provides an interesting direction for improving the Graph Attention Networks. More specifically, the authors propose a self-supervised graph attention network (SuperGAT) designed for noisy graphs. They encode positive and negative edges so that SuperGAT learns more expressive attention. They focus on two characteristics that influence the effectiveness of attention and self-supervision: homophily and average degree. They show the superiority of their method (4 variations) by comparing it with many state-of-the-art methods, in 144 synthetic datasets (with varying homophily and average node degree) and 17 real-world datasets (again with various ).


Reasons for score:
===========
Overall, I vote for accepting. I find very interesting the idea of using self-supervision to improve graph attention networks and the experiments are nicely done and convincing. The authors do an impressive work to include as much information and results as they can in the given space.


Strengths:
===========
- The paper is about an interesting problem in the ICLR community. Graph Attention Networks have gained a lot of attention in the recent years from researchers in the field of graph and node representations with applications in node classification and link prediction. The idea of adding the advancements in recent direction of self-supervised learning to improve the learnt representations seems very promising.
- The authors have done a great job in the structure and the presentation of the paper. The paper is well-written and especially the sections Experiments and Results are well-structured and contain a lot of packed details in the design and the outcome of the experiments. More specifically, Figure 4 stands out as in a very limited space contains the information for the best performed model in all 144+17 graphs!
- The contributions of this work include the proposed method (GANs with self-supervision), but also an analysis for the selection of the best model depending on two important features of the graph (homophily and average degree).
- The authors compare their method with all state-of-the-art methods and also four variations of their own model and exhaust their evaluation by testing the performance in 144 synthetic graphs and 17 real-world datasets (including the benchmark datasets that are usually being used in this domain).


Weaknesses:
===========
- The proposed method uses two known graph attention mechanism as building blocks, they use negative sampling and they add cross-entropy loss for all node labels and self-supervised graph attention losses for all layers. These building blocks and mechanisms are known in the literature, and as a result the proposed method adds incremental novelty compared to the related works.
- In Appendix, A.3, the description and discussion for t-SNE plots is limited or absent. It would be better to add more details to it, as for example why this is a good representation and how the representations improve or not based on the hyperparameters. Also, how the results in the subfigures differ in terms of representations. It is difficult to get any insights from these plots.


Questions during rebuttal:
===========
- Overall, my recommendations for more analysis and insights of the results are all responded from the Appendices. I would like a comment and clarification from the authors regarding Appendix A.3 Figure 5 (t-SNE plots), even though it is not in the main paper submission.
- My understanding is that the authors are going to release the code upon acceptance, is this correct? In the repository that the code will be released, it will be useful to also add links to all 17 public datasets to ease research in the field.

---

> ### Author Response · Authors · 2020-11-16
> **Response to R2**
>
> Thank you for your review and positive feedback. We addressed the first weakness about novelty in the [general response](https://openreview.net/forum?id=Wi5KUNlqWty&noteId=iMkhL_qmEBe). We provide answers to the second weakness and questions.
>
> > In Appendix, A.3, the description and discussion for t-SNE plots is limited or absent. It would be better to add more details to it […] Also, how the results in the subfigures differ in terms of representations. It is difficult to get any insights from these plots. [...] I would like a comment and clarification from the authors regarding Appendix A.3 Figure 5 (t-SNE plots), even though it is not in the main paper submission.
>
> Appendix A.3. is about t-SNE plots of input features of synthetic graphs, and it is not a qualitative evaluation of the learned node representations. We put in this figure to show what the synthetic datasets look like, as they are not familiar to readers in our field compared to the benchmark datasets. So we are simply showing some (small) examples of the synthetic graphs, to illustrate how the average degree and homophily vary according to $\delta$ and $p_{in}$.
>
> Sorry for the confusion. We added a more detailed description in Appendix A.3.
>
> > My understanding is that the authors are going to release the code upon acceptance, is this correct? In the repository that the code will be released, it will be useful to also add links to all 17 public datasets to ease research in the field.
>
> We have already uploaded our codes in the supplementary material. We will open our GitHub link if this paper gets accepted. This code contains links to all 17 public datasets, thanks to the open source communities in our research field: PyTorch Geometric & Open Graph Benchmark.

---

### Official Review · AnonReviewer4 · 2020-10-29
**R4**

**Rating:** 4
**Confidence:** 4

**Review:**

In this paper, the authors propose an enhanced GAT model named SuperGAT by adding link prediction as an auxiliary task when conducting node classification and compare different methods of attention forms. The authors have also conducted evaluation based on both synthetic datasets and real-world datasets to analyze how different attention methods perform on various data and task types. In general, the paper is written well and easy to follow. However, there are still several issues the authors need to address:

First, the authors need to better justify the novelty of the proposed method. The claimed self-supervision task can be considered as general link prediction task where the attention weights is used as features. Though they authors propose two new types of attention forms, i.e., 1) scaled dot-product and 2) mixed GO and DP, they are normalized and combined version of existing attention mechanisms. I suggest the authors to better justify the novelty of the proposed technique and how they differ from existing work.

Another major concern is the experiment settings. It is good to see that the authors raise several research questions to guide experiment design. However, some assumptions in these research questions are questionable. For example, in RQ1, the authors claim that “ideal node representation can be generated by aggregating only neighbors with the same label”. As the neighborhood information of a node can be also informative when predicting node labels in certain cases, a good node representation does not necessarily need to only aggregates neighbors of the same labels, which makes the proposed method that uses KL divergence to compare label-agreement and graph attention questionable. I suggest the authors to provide more justification on this assumption. RQ2’s primary goal is to understand how different graph attention methods perform for the link prediction task, it would be better if the authors can justify why they didn’t conduct experiments where only link prediction (self-supervised) loss is used and discard the node classification task. In RQ3, the authors hypothesize that “different graph attention will have different abilities to model graphs under various homophily and average degree”. This is probably true. However, given the so many graph properties (e.g., degree distribution, graph diameter, and average clustering coefficient) and model configuration (e.g., # of layers and task type), it is unclear why the authors choose these two controlled variables and why they believe they are the most important ones. I suggest the authors provide more rationale on how they choose the controlled variables and how other factors may impact model performance.

Another minor question is that why do the authors add an activation function for $e_{ij, DP}$ in Eqn. 4, given that $e_{ij, DP}$ is already a dot product that indicates the weight of a link. It would be better if the authors can elaborate more on the design rationale.

In summary, I think the authors focuses on an interesting problem but need to further address the issues listed above.

---

> ### Author Response · Authors · 2020-11-16
> **Response to R4**
>
> Thank you for your review and constructive feedback. We addressed questions about novelty and the choice of graph properties in the [general response](https://openreview.net/forum?id=Wi5KUNlqWty&noteId=iMkhL_qmEBe). We provide answers to your individual questions below.
>
> > In RQ1, the authors claim that “ideal node representation can be generated by aggregating only neighbors with the same label”. As the neighborhood information of a node can be also informative when predicting node labels in certain cases, a good node representation does not necessarily need to only aggregates neighbors of the same labels, which makes the proposed method that uses KL divergence to compare label-agreement and graph attention questionable. I suggest the authors to provide more justification on this assumption.
>
> In the node classification task, we expect that the GNN produces the same representation for nodes with the same label. Mixing neighbors with different labels has the effect of mixing representations that should be different. As theorem 1 in [3] says, if we stack infinite GAT layers, node representations in the connected component will converge to the same value. If there is an edge between nodes with different labels, it will be hard to distinguish the two corresponding labels with GAT of sufficiently many layers. Based on this theorem, although we are using finite-depth GNNs, we argue that the label-agreement distribution is a simple and effective ground-truth of graph attention.
>
> Please see the red text in RQ1 on Page 4 of the revised paper.
>
> [3] Wang, Guangtao, et al. "Improving graph attention networks with large margin-based constraints." arXiv preprint arXiv:1910.11945 (2019).
>
> > RQ2’s primary goal is to understand how different graph attention methods perform for the link prediction task, it would be better if the authors can justify why they didn’t conduct experiments where only link prediction (self-supervised) loss is used and discard the node classification task.
>
> SuperGAT is focused on the node classification task, in keeping with the focus of the GAT. SuperGAT uses link prediction experiments to understand how attention learns the relational importance from edges for the node classification, but not for the link prediction itself.
>
> Figure 3 in the revised paper shows the results of changing the mixing coefficient $\lambda_E$. These results show a consistent pattern of significant decrease in the node classification performance as the mixing coefficient increases, which is equivalent to giving a large weight to the link prediction task. So we can extrapolate that when only the link prediction loss is used, the representation learned from the model would be quite poor.
>
> > Another minor question is that why do the authors add an activation function for eij,DP in Eqn. 4, given that eij,DP is already a dot product that indicates the weight of a link. It would be better if the authors can elaborate more on the design rationale
>
> SuperGAT MX (Eq. 4) multiplies GO and DP attention with the sigmoid. The motivation of this form comes from gating mechanisms such as GRU. Since DP attention with the sigmoid represents the probability of an edge, it can softly drop neighbors that are not likely linked while implicitly assigning importance to the remaining nodes. This also prevents the attention value from being highly dependent on DP, the variance of which is larger than those of GO.
>
> We revised this part to be more clear.

---

### Author Response · Authors · 2020-11-16
**General Response**

We thank the reviewers for insightful and constructive feedback. We first answer two issues raised by multiple reviewers: novelty of our models and justification of focusing on two graph properties.

### Contributions and Novelty

R1, R2, and R4 raised an issue of limited novelty of SuperGAT.

Our model improves graph attention by employing link prediction for self-supervision. Each of these building blocks is not novel, but combining them to give self-supervision to attention from edges, is not trivial. This is because simply adding the self-supervision task to the graph attention mechanism does not produce better results (RQ 1-2), and the resulting model performance varies across datasets (RQ 3-4). In addition to the model, this paper presents insightful findings of when and how to use self-supervision for graph attention. We believe these in-depth analyses and insights are important contributions to the community of researchers and practitioners working with graph neural nets. We hope the reviewers will support this view.

### Focusing on Avg. Degree and Homophily

R1 and R4 asked about our choice of two graph properties: the average degree and homophily, among many graph properties (e.g., diameter, degree sequence, degree distribution, average clustering coefficient).

There are many properties that can characterize graphs. Among them, we choose average degree and homophily as the main variables of our study because they determine the quality and quantity of labels in our self-supervised task (i.e., link prediction). Label quality and quantity directly impact the result of supervised learning [1]. Below is a sentence from our main paper, section 4, RQ3.

“From the perspective of supervised learning of graph attention with edge labels, the quality of the result depends on how noisy labels are (i.e., how low the homophily is) and how many labels exist (i.e., how high the average degree is).”

We revised the explanation under RQ3 to address this concern and clarify our reasoning. Please see the sentences in red on Page 5 of the revised paper.

There are also practical reasons to use the average degree and homophily. In RQ3, we experiment with various synthetic graphs in the space created by two properties and map the result of real-world graphs. There are three considerations in choosing graph properties for this experiment. First, the graph property can be computed efficiently even for large graphs. Second, there should be an algorithm that can generate graphs by controlling the property of interest only. Third, the property should be a scalar value because if the synthetic graph space is too wide, it would be impossible to conduct an experiment with sufficient coverage. Average degree and homophily satisfy the above conditions and are suitable for our experiment, unlike some of the other graph properties.

Upon reading the reviews, we analyzed two scalar properties, diameter and average clustering coefficient, which reviewers explicitly mentioned, but we could not find a meaningful pattern for the best graph attention model in the space of diameter and average clustering coefficient. In future work, we will explore and analyze some of the other graph properties.

[1] David Rolnick, Andreas Veit, Serge Belongie, and Nir Shavit. Deep learning is robust to massive label noise. arXiv preprint arXiv:1705.10694, 2017.

---

### Author Response · Authors · 2020-11-23
**Summary of Revision**

To all readers, we uploaded the revised paper last week based on reviewers' comments. The revised parts are highlighted in red. In summary, we added the following:
- Figure 1 to illustrate attention mechanisms of all SuperGAT variants and the original GAT (Veličković et al., 2018)
- Design rationale of SuperGAT MX
-  Explanation about the sparsity of real-world graphs in response to the limitation of our model for dense graphs
- Justification of our choice of label-agreement as ground-truth of importance using theoretical analysis of deep GATs (Wang et al., 2019)
- More details about why we chose the average degree and homophily as the main variables of RQ3 experiments
- More clear description of t-SNE plots of input graphs in appendix A.3

Again, thank you for the constructive feedback. If there is an issue that has not been addressed, we would be happy to discuss it.

---

### Decision · Program_Chairs · 2021-01-07
**Final Decision**

**Decision:**

Accept (Poster)

**Comment:**

Two reviewers are very positive about this paper and recommend acceptance, one indicates rejection and one is on the fence. Although all referees appreciate the extensive experiments and analysis presented in the paper, their main concerns are related to the limited superiority of the method wrt state of the art [R1], seemingly arbitrary choices and questionable assumptions [R4]. The rebuttal adequately addresses R1's concerns by highlighting statistical significance of the results, and partially covers R4's concerns. Although the proposed approach may be perceived as incremental [R1, R2, R3, R4], the authors argue that introducing self-supervision to graph attention is not trivial, and emphasize their findings on how/when this is beneficial. Moreover, R2 and R3 acknowledge that the contribution of the paper holds promise, is worth exploring, and may be useful to the research community. Most reviewers are satisfied with the answers in the rebuttal. After discussion, three referees lean towards acceptance and the fourth reviewer does not oppose the decision. I agree with their assessment and therefore recommend acceptance. Please do include your comments regarding the choice of average degree and homophily in the final version of paper.